# Saharan Dust Impact on Radiative Heating Rate Errors Inherent in Reanalysis Data in the African Easterly Wave Development Region

Ruby W. Burgess [1] and Mayra I. Oyola-Merced [1]

[1]University of Wisconsin-Madison, Madison, WI 53706, USA

**Correspondence:** Ruby W. Burgess  (rwburgess@wisc.edu)

**Abstract.** This study thoroughly examines the impact of aerosols on atmospheric heating rates over the North Atlantic Ocean, with a specific focus on developing African Easterly Waves (AEWs). The analysis leverages data from the NASA DC-8 aircraft, dropsonde profiles, lidar observations, and satellite-based precipitation data obtained during NASA's CPEX-CV field campaign, as well as the MERRA-2 and CAMS reanalyses. Using a four-stream radiative transfer model, the research focuses specifically on days characterized by Saharan dust coinciding with a developing AEW and contrasts its findings with a notable dust-only event in June 2020. The findings reveal notable differences in shortwave (SW) heating rates of over 1.5 K/day between reanalysis and observations, underlining the persistent challenges in accurately representing aerosol effects in the atmosphere, even after assimilating observational data. These discrepancies were present on days with both background and high dust concentrations, emphasizing the challenges in accurately representing aerosol radiative effects in models and highlighting the urgent need for improved aerosol representation in reanalysis datasets. Differences in heating rates were analyzed in a case study of two developing AEWs, one leading to a Category 4 Hurricane (Fiona) and another leading to a short-lived tropical storm (TS Hermine).

## 1   Introduction

Over the past two decades, substantial advancements have been made in characterizing aerosol properties, as well as in identifying their spatiotemporal distribution and their influence on the planet's radiative equilibrium (Ramanathan et al., 2001). This research has culminated in the recognition that aerosols have both a "direct effect" on climate by altering the Earth's radiative budget and redistributing heat throughout the atmosphere, as well as an "indirect effect" by impacting cloud formation, precipitation, and optical properties (IPCC, 2023). These effects are contingent on the concentration and altitude of aerosols (Lyapustin et al., 2011; Bauer and Menon, 2012; Xu et al., 2017). In the same period, significant strides have been made in aerosol modeling, data assimilation techniques for Numerical Weather Predictions (NWP) applications, and the development of precise 3-D aerosol models. These developments have enabled a more accurate representation of aerosols in weather models and reanalysis, leading to improvements in forecast accuracy (Mulcahy et al., 2014; Toll et al., 2016). Furthermore, these advancements have opened new avenues for advanced research on aerosol effects and provided the potential for monitoring air quality events. Nevertheless, uncertainties persist, especially concerning the atmosphere's response to various physical properties of aerosols, particularly on daily timescales that affect weather patterns (Mulcahy et al., 2014; Toll et al., 2016; Zhang

et al., 2016). This is due to significant limitations in accurately characterizing aerosols, which are crucial for forecasting and understanding the evolution of weather systems and processes.

Aerosols, with characteristics such as concentration, size distribution, composition, vertical distribution, hygroscopicity, and mixing state, dynamically influence heating rates in the Earth's atmosphere. This influence stems from their complex role in orchestrating radiative processes within large-scale weather systems. For example, aerosol concentration significantly dictates the scattering and absorption of solar radiation, leading to regional variations in heating rates. Similarly, size distribution of aerosol particles governs their efficacy in scattering or absorbing radiation, impacting temperature gradients and atmospheric stability. Composition is pivotal, inducing localized atmospheric heating or cooling. Vertical distribution intricately shapes aerosol radiative effects across distinct atmospheric levels. Hygroscopic properties alter aerosol optical characteristics as they interact with water vapor, and mixing state complicates their radiative consequences. Therefore, understanding these aerosol-induced changes in heating rates is crucial for enhancing the accuracy of weather forecasting models and the reliability of reanalysis data. This knowledge enables a more precise representation of atmospheric processes and the development of weather systems.

The North Atlantic basin provides the setting for these processes to coexist. On a protagonist role are the African Easterly Waves (AEWs, Burpee, 1972; Reed et al., 1988; Thorncroft and Blackburn, 1999). Along with the African Easterly Jet (AEJ), they are the primary triggers of regional and synoptic weather events over the Atlantic basin (Reed et al., 1977), having devastating societal consequences over Africa, the Caribbean and the United States. Studies show, for example, that the AEJ-AEW system influences convection and rainfall over West Africa (Carlson, 1969; Reed et al., 1977), while more than half of the tropical cyclones (TCs) that have been observed to develop over the eastern Atlantic Ocean have AEW origins (Landsea et al., 1998). Another important phenomenon is the Saharan dust (and associated Saharan Air Layer, SAL), a prominent aerosol feature that covers a vast portion of the Atlantic Ocean during boreal Spring and Summer (Carlson and Prospero, 1972; Dunion and Velden, 2004). The Saharan dust is believed to alter both short wave (SW) and longwave/infrared (LW/IR) solar radiation (Dunion and Velden, 2004) as well as temperatures at the surface and aloft (Nalli and Stowe, 2002; Oyola, 2015), decrease vertical wind shear, induce thermodynamic stability, and most notably, influence the genesis of tropical storms (TS) and hurricanes (Dunion and Velden, 2004; Pratt and Evans, 2009). Given that they share similarities in seasonality and geographical extent, the AEWs and Saharan dust are consequently coupled to influence each other: on larger timescales, processes like the AEW trigger Saharan dust lofting by enhancing diurnal emission mechanisms (Dunion and Velden, 2004). On the other hand, dust atmospheric feedbacks influence the AEWs through direct and indirect radiative effects (Grogan and Thorncroft, 2019). The United States Geostationary Operational Environmental Satellites (GOES) satellite imagery in Fig. 1 depicts an example of these interactions. We have advanced our understanding of how Saharan dust affects AEW's structure and evolution, however, despite more than two decades of studies, no conclusive evidence has been agreed upon. From the observational/reanalysis standpoint, most studies examining the effects of Saharan dust on AEWs using analytical approaches have produced contradictory results, and/or have only focused on attributing AEW growth or decay to dust-induced changes in the static stability alone (e.g. Karyampudi et al., 1999; Jones et al., 2004; Reale et al., 2009; Jury and Santiago, 2010; Reale et al., 2011; Ma et al., 2012). A significant limitation with these studies, and a possible reason behind their discrepancies, is that most focus on total column aerosol loading (or AOD) but have failed to properly address the relationship between the AEWs

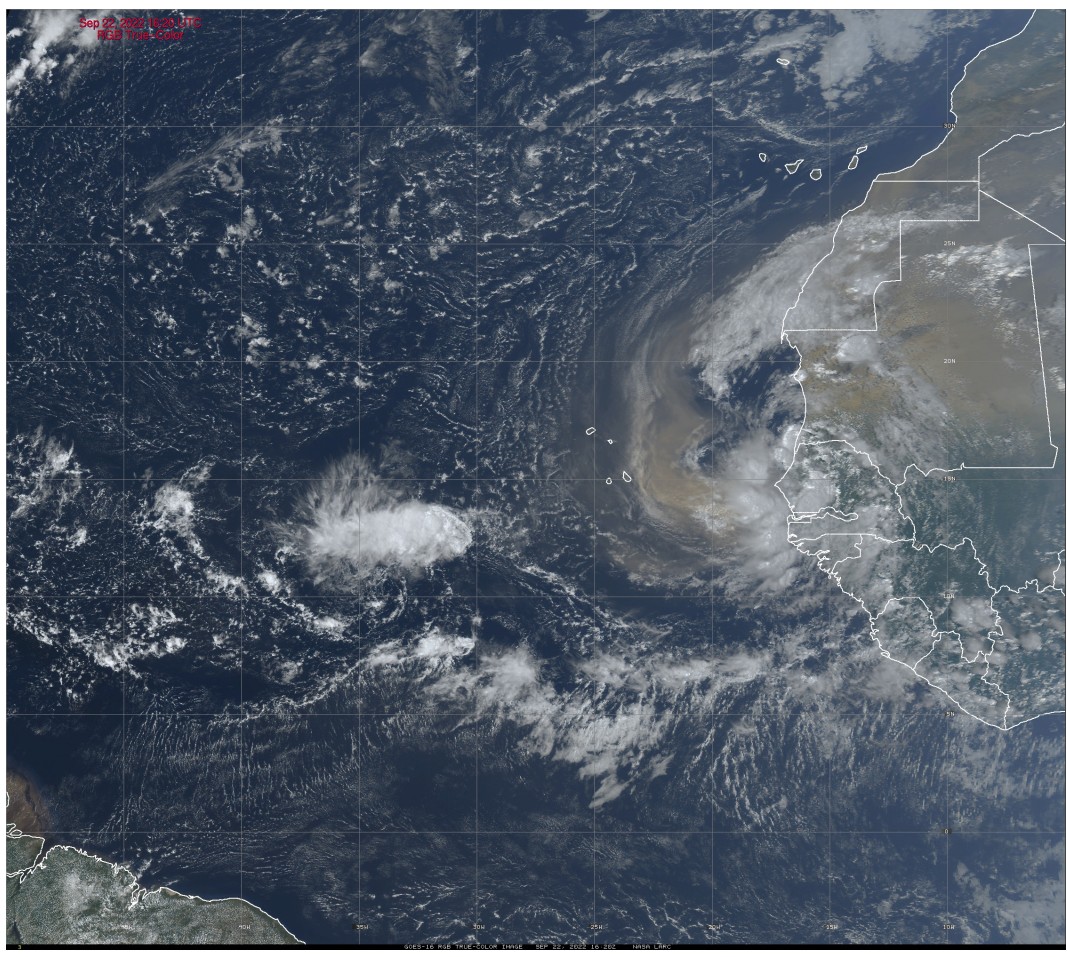

**Figure 1.** Dust interacting with an African Easterly Wave on 22 September 2022 during the CPEX-CV field campaign as captured by GOES-16 (GOES-R Series Program, 2019).

and changes in the vertical distribution of dust aerosols. Also, these studies do not directly address the dust impact at different stages of the AEW, nor clearly discretize changes in airmass in both systems with transatlantic passage. Although it seems a straightforward issue to address, previous research has been limited because retrieving good-quality measurements of vertical profiles of atmospheric parameters (moisture, wind, temperature, and vertical information of aerosols) in Saharan dust events remains an extremely challenging endeavor, particularly in the lower atmosphere, as most of these techniques are biased due to changes in atmospheric composition itself (Nalli et al., 2011; Oyola, 2015).

There is an opportunity to characterize the impact of dust aerosols in AEWs, aiming to enhance our comprehension of their feedback mechanisms. This can be achieved by leveraging radiative transfer, aerosol vertical data from reanalysis, and, when accessible, utilizing existing field campaign datasets for atmospheric closure experiments. Obtaining high-resolution

aerosol profiles from observations poses a significant challenge, particularly over oceanic regions. However, when such data is accessible, it can contribute significantly to addressing some of the questions that remain unanswered.

In this study, we shed light on the importance of accounting for the vertical distribution of Saharan dust in the context of AEW development by showing the impact of anomalous dust loadings on modifying atmospheric heating rates at levels critical to AEW development (1000-500 hPa). We show that these impacts are not well captured by reanalysis, a key factor in improving the modeling of AEW development. We integrate data collected during the airborne National Aeronautics and Space Administration (NASA) Convective Processes Experiment – Cabo Verde (CPEX-CV) and profiles from two different global reanalyses into a four-stream radiative transfer model. We examine radiative heating rates within Saharan dust plumes associated with AEWs during the intensive observation period (IOP). The CPEX-CV datasets provide a distinctive opportunity for this model evaluation, featuring collocated aircraft instrumentation that simultaneously measured high-resolution vertical aerosol profiles and atmospheric profiles via dropsondes over AEWs. Utilizing this in situ instrumentation, our goal is to assess the reanalyses' accuracy in depicting aerosol radiative properties. Specifically, within the observational constraints of the limited dataset available for this study, our objectives include quantifying the magnitude of aerosol-induced heating rates—particularly those associated with high dust loading—and shedding light on their potential to influence model representation of AEW development.

## 2    Data and methods

The analysis was conducted over the North Atlantic Ocean, in a box spanning 0°to 25°N and 15°W to 35°W to the west of the Sahara Desert. Data from several reanalyses and the CPEX-CV field campaign were used to analyze the effects of aerosols on atmospheric profiles and their role in the development of AEWs.

### 2.1    CPEX-CV

CPEX-CV was conducted between 1 and 30 September 2022 out of Cabo Verde over the North Atlantic. Its objectives included examining the interplay of atmospheric dynamics, properties of the marine boundary layer, convection, the Saharan Air Layer and Saharan dust, and their interactions at different spatial scales. The mission aimed to enhance our comprehension and predictive capabilities regarding the lifecycles of processes such as AEWs, aiming to increase our understanding of such processes in a data-scarce region such as the tropical East Atlantic. During the field campaign, data were collected during 13 research flights from the NASA DC-8 aircraft by several instruments including the ones described below. We make use of data from the seven research flights that coincided with a developing AEW.

#### 2.1.1    AVAPS dropsondes

The Advanced Vertical Atmospheric Profiling System or AVAPS (Hock, 1999), is a dropsonde system providing vertical profiles of pressure, temperature, specific humidity, and winds that was used onboard the DC-8 during CPEX-CV. Dropsondes were launched at multiple locations during each flight. The profile altitude was limited to the DC-8 aircraft's maximum altitude

of 42,000 ft, and most profiles did not contain data above 200 hPa. We employ 64 dropsonde profiles of pressure, temperature, and specific humidity throughout seven research flights to characterize atmospheric conditions in our analysis. We use mean daily profiles of pressure, temperature, and specific humidity from the AVAPS dropsonde dataset to calculate mean heating rates for two days of interest (09 and 22 September).

### 2.1.2 HALO

The NASA Langley High Altitude Lidar Observatory or HALO (Bedka et al., 2021) is a lidar system operated from an airborne platform to provide nadir-viewing profiles of water vapor, methane columns, and profiles of aerosol and cloud optical properties. The HALO profiled the vertical distribution of aerosol in the atmosphere during each of the research flights used in our analysis. The 532 nm aerosol extinction coefficient, inferred from the aerosol backscatter (Carroll et al., 2022; Lei et al., 2022; Lenhardt et al., 2022) is used in our experiments as a measure of extinction coefficient. The latitude and longitude data from the HALO dataset were used to determine the flight track location for each flight used in the analysis.

### 2.2 MERRA-2

The Modern-Era Retrospective Analysis for Research and Applications, Version 2 (MERRA-2, Buchard et al., 2017; Gelaro et al., 2017; Randles et al., 2017) is a reanalysis dataset developed by NASA that provides comprehensive and high-quality atmospheric data from 1980 onward, including the assimilation of aerosols and a representation of their interactions with other physical processes. We utilize the 3D 6-hourly Analyzed Meteorological Fields dataset (or inst6_3d_ana_Nv on 72 levels) for profiles of pressure, temperature, specific humidity, and ozone mixing ratio. For aerosol, we utilize the inst3_3d_aer_Nv collection, which includes instantaneous 3-dimensional 3-hourly data within MERRA-2. This dataset encompasses assimilated aerosol mixing ratio parameters at 72 model layers, including dust, sulfur dioxide, sea salt, black carbon, and organic carbon. Similarly, we also obtain 3-hourly Aerosol Optical Depth (AOD) Analysis from the inst3_2d_gas_Nx.

Additional treatment is required to be able to obtain extinction coefficient profiles from dust concentration. We calculate volume extinction coefficient at each level from dust mixing ratio for each of the five size-bins provided in the aerosol mixing ratio dataset, using the following equation:

$$\beta_e = k_e \rho_{air} = \frac{3 R_{DU} Q_{ext}}{4 r \rho_p} \rho_{air} \tag{1}$$

where $k_e$ is the mass extinction coefficient in m$^2$ kg$^{-1}$, $\rho_{air}$ is the air density in kg m$^{-3}$, $R_{DU}$ is the dust mass mixing ratio for a specific bin in kg kg$^{-1}$, $Q_{ext}$ is the extinction efficiency, $r$ is the particle radius in m, and $\rho_p$ is the particle density in kg m$^{-3}$. The air density was provided by the MERRA-2 analyzed meteorological fields. The particle radius used for each of the five size-bins is 0.73 $\mu m$, 1.4 $\mu m$, 2.4 $\mu m$, 4.5 $\mu m$, 8.0 $\mu m$ respectively. The particle density is 2500 kg m$^{-3}$ for particles of mean radius of 0.73 $\mu m$, and 2650 kg m$^{-3}$ for the rest of the size-bins (GMAO, 2023). The extinction efficiency was approximated for each size-bin using values from the Goddard Chemistry Aerosol Radiation and Transport (GOCART) module (GMAO, 2023) that correspond to the closest particle radius for each bin. The HALO data collected during the CPEX-CV campaign

were assimilated into the MERRA-2 reanalysis (Nowottnick et al., 2023), and our analysis sheds light on the performance of the assimilation.

## 2.3 CAMS

Because the CPEX-CV data were assimilated into the MERRA-2 reanalysis dataset used in this study, we use the Copernicus Atmosphere Monitoring Service reanalysis (CAMS, Inness et al., 2019), which did not assimilate data from CPEX-CV, as a reference to assess the impacts of assimilation on the reanalysis. CAMS is a reanalysis dataset that comprises 3D time-consistent atmospheric composition fields, including aerosols, chemical species, and greenhouse gases. We utilize the 3-hourly datasets on 25 pressure levels for temperature, specific humidity, and dust aerosol mixing ratio at three different particle size ranges (0.03 - 0.55 $\mu$m, 0.55 - 0.9 $\mu$m, 0.9 - 20 $\mu$m), as well as the total column AOD at 550 nm. Similar to the MERRA-2 dataset, we calculate the extinction coefficient at each level for the 3 dust size-bins listed above using the following formula:

$$\beta_e = k_e \rho_{air} = \frac{3 R_{DU} Q_{ext}}{4 r \rho_p} \cdot \frac{p}{R_d T_v} \tag{2}$$

where $T_v = (1 + 0.61q)T$ and $p$ is the pressure in hPa, $R_d$ is the gas constant for dry air in J kg$^{-1}$ K$^{-1}$, $T_v$ is the virtual temperature in K, $q$ is the specific humidity in kg kg$^{-1}$, and $T$ is the temperature in K. The pressure, specific humidity, and temperature were provided by the CAMS dataset. Since CAMS also uses GOCART aerosol properties, the values for extinction efficiency, particle radius, and particle density for each of the three size-bins are the same used for MERRA-2 for particle radii sizes of 0.24 $\mu m$, 0.8 $\mu m$, and 8 $\mu m$ respectively. Similarly to the MERRA-2 dataset, the values for extinction coefficient were added together to calculate the total dust aerosol extinction coefficient. Because each size-bin represents a range of particle sizes and the extinction efficiency depends on particle size, the accuracy of the extinction coefficient remains limited for both the MERRA-2 and CAMS datasets.

## 2.4 AEW Tracking

We use the AEW tracker described in Lawton et al. (2022) to track the center of several AEWs of interest. The tracker calculates curvature vorticity at 700 hPa using the nondivergent component of the 700-hPa wind averaged within a radius of 600 km of each grid point. We use the positional dataset which supplies an approximation of the location of the center of the AEW at a 6-hour time step to collocate the center of the AEW with the nearest MERRA-2 and CAMS reanalysis datasets.

## 2.5 Data processing

### 2.5.1 Observational analysis

The Integrated Multi-satellitE Retrievals for Global Precipitation Measurement (IMERG, Huffman et al., 2020) is a dataset developed and provided by NASA that offers global precipitation data by merging and integrating data from the Global Precipitation Measurement (GPM) satellite constellation.

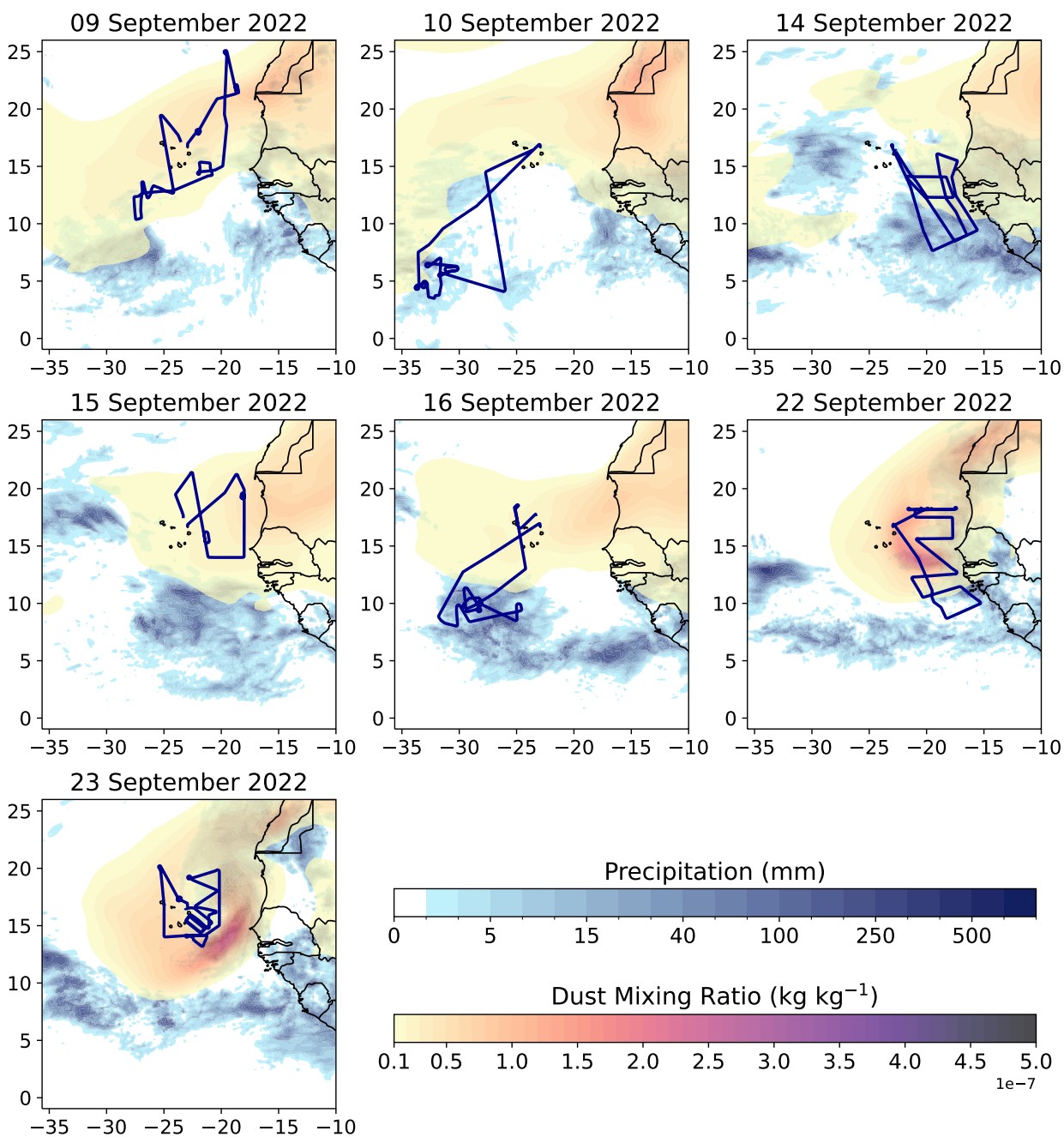

**Figure 2.** Flight tracks (blue) with overlaid total dust mixing ratio from MERRA-2 and daily accumulated precipitation from IMERG for seven of the research flights during the CPEX-CV field campaign.

We leverage the precipitation data within the IMERG dataset to identify and follow the progression of the sampled AEWs within our designated geographic region. To explore the dynamics of dust in relation to convection and precipitation, we superimpose IMERG daily accumulated precipitation with the MERRA-2 total dust mixing ratio, enabling the creation of daily maps (e.g. Fig. 2). The MERRA-2 total dust mixing ratio was derived by aggregating the contributions from five distinct particle size-bins. These maps allow us to explore how and where dust aerosols may be influencing AEW dynamics, providing a more comprehensive view of the factors affecting AEW behavior. In the process of our analysis, we leveraged these maps, in conjunction with the daily forecast reports from CPEX-CV to identify specific days of interest. Furthermore, we utilized these maps to track the temporal evolution of dust concentration throughout the field campaign. Additionally, we assessed the integration of CPEX-CV data into the MERRA-2 reanalysis, examining how this assimilation impacted the overall dataset.

### 2.5.2 Data processing for radiative transfer

We use the positional data from the CPEX-CV HALO dataset to collocate data from the two reanalyses (MERRA-2 and CAMS) with the CPEX-CV dataset. We process the AVAPS dropsonde data to select profiles of pressure, temperature, and specific humidity with sufficient information to be run in the Fu-Liou-Gu (FLG) radiative transfer model (RTM) (Gu et al., 2011). We generate profiles from MERRA-2 and CAMS reanalysis at each dropsonde profile location. Similarly, we process extinction coefficient profiles from HALO to select profiles providing sufficient aerosol extinction coefficient data. We select extinction coefficient profiles from the HALO, MERRA-2, and CAMS datasets matching the location of the selected dropsonde profiles. We interpolate all profiles to 72 vertical levels, restricting both the atmospheric and extinction coefficient profiles to below the 100 hPa level. Values of AOD are retrieved for each location using the 532 nm Total Optical Thickness from CPEX-CV, Aerosol Optical Depth Analysis from MERRA-2 and Total Aerosol Optical Depth at 550 nm from CAMS. These profiles, along with AOD values, are used for the calculation of heating rates using the FLG RTM. A summary of datasets used as inputs in the FLG RTM is given in Table 1. For our AEW case study, we calculate a mean pressure, temperature and specific humidity profile from the AVAPS dropsonde dataset and a mean extinction coefficient profile from the HALO extinction coefficient dataset for each flight. We generate a corresponding mean atmospheric profile and extinction coefficient profile from both reanalyses using a collocated dataset. These mean profiles are used in the case study of daily mean heating rates during the AEW events on 09 September and 22 September. We also use the AEW tracking methodology described in section 2.4 to generate heating rate profiles at 6-hour time steps during the development of the AEWs studied.

### 2.6 Fu-Liou-Gu Radiative Transfer Model

The Fu-Liou-Gu RTM calculates heating rates and irradiances (fluxes) from profiles generated from the datasets described above. The FLG RT scheme, as proposed by Gu et al. (2011), represents an upgraded iteration of the RTM originally developed by Fu and Liou in 1992 and 1993 (Fu and Liou, 1992, 1993). This refined model offers improved parameterizations for aerosol properties, which enable more accurate simulation of radiative effects, aligning more closely to real-world observations. The delta-four-stream approximation is utilized for solar radiative flux calculations (Liou et al., 1988) and the delta-two-and-four-stream approximation is employed for LW/IR radiative flux calculations (Fu et al., 1997) in the model. The model divides the

**Table 1.** FLG Input parameter datasets for PTQ (atmospheric profile of pressure, temperature, and humidity) and extinction coefficient (see text for further details on parameter calculations).

| Dataset | PTQ | Extinction coefficient |
|---------|-----|------------------------|
| CPEX-CV | AVAPS Dropsondes | HALO Extinction Coefficient |
| MERRA-2 | 6-hourly Analyzed Meteorological Fields | Extinction coefficient from 3-hourly Dust Mixing Ratio |
| CAMS | 3-hourly pressure, temperature, specific humidity | Extinction coefficient from 3-hourly Dust Mixing Ratio |

solar and LW/IR spectra into 6 and 12 bands respectively, determined by the locations of absorption bands, and the calculations
incorporate the effect of absorption by the H2O continuum and various minor absorbers within the solar spectrum in addition
to the principal absorbing gases.

### 2.6.1   OPAC

The current FLG radiation scheme contains a total of 18 aerosol types parametrized by the Optical Properties of Aerosols and
Clouds (OPAC, Hess et al., 1998) database. This database provides humidity-aware single-scattering properties for spherical
aerosols computed from Lorenz Mie theory, for 60 wavelengths in the spectral region between 0.3 $\mu$m and 40 $\mu$m. These 60
bands are interpolated into the 18 bands of the FLG RT scheme. The 18 types of aerosol include maritime, continental, urban,
five size-bins for mineral dust, insoluble, water soluble, soot (BC), sulfate droplets, sea salt in two modes (accumulation and
coarse mode), and mineral dust in four different modes (nucleation, accumulation, coarse, and transported mode). For the
purposes of this study, we employ the mineral dust transported mode.
We acknowledge that more advanced aerosol climatologies are available today compared to OPAC. However, OPAC remains
widely used in most NWP models, which is why it was chosen for this study, given both of the reanalysis also use it. The choice
was made to ensure consistency with the models that are most commonly used in the community. We also recognize that there
are several issues related to aerosol properties in OPAC, such as uncertainties in AOD, hygroscopic growth, particle size
distribution, and refractive indices, which have been documented in previous publications (e.g., Dubovik et al., 2002; Kahn
et al., 2005; Levy et al., 2010). Similarly, we recognize potential uncertainties associated with the conversion of backscatter to
extinction in the lidar observations. However, it is important to note that these conversions are performed by the science team
responsible for the data and as, users, we have assumed that these conversions are accurate to the best possible knowledge of
the science team. However, we recognize that these assumptions may introduce uncertainties, particularly in how they might
propagate into the calculated heating rate profiles.
Previous studies have highlighted these issues. For example, Dubovik et al. (2002) examine the influence of particle non-
sphericity on the retrieval of aerosol optical properties, while Kahn et al. (2005) discuss the uncertainties in aerosol models
derived from satellite data. Additionally, Levy et al. (2010) address the challenges in retrieving accurate aerosol optical proper-
ties and their implications for climate models. These studies suggest that the propagation of such uncertainties can significantly
affect the accuracy of radiative transfer models, highlighting the importance of ongoing evaluation and refinement in this area.

 **2.6.2 Calculation of heating rates**

Following a similar approach to Oyola et al. (2019), we run the FLG RTM ingesting atmospheric profiles from the three datasets (MERRA-2, CAMS, and CPEX-CV) to retrieve heating rates throughout the vertical layer at each of the selected dropsonde locations. Simulations are performed after accounting for the solar zenith angle at the corresponding local time and location. The heating rate at each levels is described by (e.g. Petty, 2008):

$$
\mathcal{H}(z) \equiv -\frac{1}{\rho(z)C_p}\left\{ -[F_i^\uparrow(0) - \Delta\tilde{v}_l\pi\bar{B}_l(z)]\frac{\partial\tau_i(0,z)}{\partial z} \right.
$$

$$
-[F_i^\downarrow(\infty) - \Delta\tilde{v}_l\pi\bar{B}_l(z)]\frac{\partial\tau_i(z,\infty)}{\partial z}
$$

$$
-\Delta\tilde{v}_l\pi\int_z^\infty[\bar{B}_l(z') - \bar{B}_l(z)]\frac{\partial^2\tau_i(z,z')}{\partial z'dz}dz'
$$

$$
\left. -\Delta\tilde{v}_l\pi\int_0^z[\bar{B}_l(z') - \bar{B}_l(z)]\frac{\partial^2\tau_i(z',z)}{\partial z'dz}dz' \right\} \tag{3}
$$

where $\rho(z)$ is the air density at level z, $C_p$= 1005 J kg$^{-1}$ K$^{-1}$ is the specific heat capacity of air at constant pressure, $\tau_i$ is the band average flux transmittance, $\tilde{v}_l$ represents the spectral interval or band (SW, LW/IR), $F_i^\uparrow$, $F_i^\downarrow$, $F_i^\uparrow(0)$, $F_i^\downarrow(\infty)$ are fluxes where the arrows represent the direction of incoming flow (from surface up from top of atmosphere to surface), and the indices 0 and $\infty$ represent the surface and TOA respectively. The heating rate is dominated in magnitude by the first two terms: the first term quantifies radiative exchange with the boundary layer and is generally a heating term, while the second term quantifies radiative exchange with the top of the atmosphere and thus predicts longwave/infrared cooling to space. In its summarized form, the heating rate equation can be stated as:

$$
\mathcal{H}(z) \equiv -\frac{1}{\rho(z)C_p}\frac{\partial F_{net}}{\partial z} \tag{4}
$$

Here, $F_{net}$ is the net flux given by the difference between upward and downward-directed fluxes. We set a control RTM run for each profile where no aerosol feedback is included, which we run parallel to the RTM run using the extinction coefficient profiles calculated from each of the three respective datasets for each of the profile locations.

**2.6.3 Heating rate experiments**

We distinguish between background AOD and anomalous AOD to emphasize the impact of dust concentration on atmospheric heating rates. We define a threshold of AOD < 0.2 (which is considered a background AOD level) calculated from the CPEX-CV 532 nm total optical thickness to select background dust concentration profiles. We obtain 32 dropsonde locations which fit the condition of background AOD (AOD < 0.2), and we select profiles from all three datasets at these locations. We then select the top 32 dropsonde locations with highest AOD calculated from CPEX-CV 532 nm total optical thickness and define these as anomalous dust concentration profiles, and select profiles from all three datasets at these locations. The resulting mean AOD value for the background and anomalous cases for each dataset are shown in Table 2. The profiles selected for

**Table 2.** Mean AOD for background and anomalous cases over the seven research flights for CPEX-CV, MERRA-2 and CAMS.

| Dataset | Mean Background AOD | Mean Anomalous AOD |
|---------|---------------------|--------------------|
| CPEX-CV | 0.09 | 0.83 |
| MERRA-2 | 0.22 | 0.46 |
| CAMS | 0.27 | 0.33 |

the background and anomalous cases for MERRA-2 and CAMS are based on the CPEX-CV AOD threshold, not MERRA-2 and CAMS AOD values, and thus have a differing range of AOD values. The 64 dropsonde profiles of temperature, specific humidity, and extinction coefficient are ingested into the FLG RTM, which is run for all three datasets (CPEX-CV, MERRA-2, and CAMS) at each dropsonde location at the time of launch. We refer to these runs as the aerosol-aware case. A control run where the aerosol parameter was turned off is also performed for each run. The mean shortwave (SW), longwave/infrared

(LW/IR), and total heating rate differences between the aerosol runs and the control runs are calculated using the FLG RTM. The results are plotted for background dust concentration profiles in Fig. 4 and for anomalous dust concentration profiles in Fig. 5.

## 3   Results and discussion

### 3.1   Description of AEW events during CPEX-CV

The analysis was focused on data collected within a region defined by latitudes ranging from $0°$ to $25°$ N and longitudes from $15°$ to $35°$ W. During the CPEX-CV field campaign, the 13 DC-8 research flights sampled 10 different African Easterly Waves (AEWs) identified as AEW 1 through AEW 10. Four of these waves developed into named TSs (AEW 4, 5, 6, 8), with two intensifying into hurricanes (AEW 4, 6). For this study, we utilized profiles collocated with dropsondes obtained during developing AEW events, which correspond to the flights on 9, 10, 14, 15, 16, 22, and 23 September, resulting in 64 profiles over

4 AEWs or their surrounding environments. Table 3 provides specific information about the developing AEWs corresponding to the research flights. Environmental conditions varied for each flight, as illustrated in Fig. 2, which displays flight tracks for several days of interest and corresponding weather conditions. Integrated dust concentration from MERRA-2 and accumulated IR-MW precipitation from IMERG highlight different regimes sampled, such as conditions where mainly dust is present (09 and 15 September), conditions where the "dusty" outer environment of the AEW was sampled (14 September), and situations

where AEWs interacted with heavy dust loadings (22-23 September).

On 09 September, AEW 4 was located off the west coast of Africa, later evolving into TS Fiona on 14 September and further intensifying into a hurricane on 18 September. Fiona reached Category 4 with highest 1-minute sustained winds of 140 mph (220 km/h) and produced catastrophic damage to many islands in the Caribbean. On 23 September, it transitioned into an extra-tropical cyclone, directly impacting the Atlantic portion of Canada and becoming the costliest cyclone in Canadian

**Table 3.** Flight date, time, location, mean and maximum AOD, location relative to AEW and corresponding tropical cyclone for seven days corresponding to a developing AEW.

| Flight Date | Flight times | Location | Mean AOD | Max AOD | Location Relative to AEW | Corresponding TC |
|---|---|---|---|---|---|---|
| 09 September | 12:06:35 to 20:40:03 UTC | 10.4° – 25.1°N, 18.5°W – 27.6°W | 0.25 | 1.69 | In AEW 4 region | Fiona |
| 10 September | 13:42:27 to 21:19:37 UTC | 3.5° – 16.9°N, 22.9°W – 33.8°W | 0.07 | 0.26 | In AEW 4 region | Fiona |
| 14 September | 09:03:44 to 16:27:09 UTC | 7.7° – 16.9°N, 15.6°W – 23.0°W | 0.06 | 1.02 | Between AEW 5 and AEW 6 | Gaston and Ian |
| 15 September | 15:04:47 to 20:26:22 UTC | 14.0° – 21.4°N, 18.0°W – 24.0°W | 0.36 | 1.10 | North of AEW 6 | Ian |
| 16 September | 12:52:20 to 20:26:15 UTC | 8.0° – 18.6°N, 22.9°W – 31.8°W | 0.23 | 1.14 | In AEW 6 region | Ian |
| 22 September | 04:53:21 to 12:36:42 UTC | 8.7° – 18.4°N, 15.3°W – 23.0°W | 1.02 | 3.34 | In AEW 8 region | Hermine |
| 23 September | 06:48:29 to 14:17:34 UTC | 13.2° – 20.2°N, 20.2°W – 25.5°W | 0.52 | 1.51 | In AEW 8 region | Hermine |

history. It finally dissipated on 27 September 2022. The DC-8 aircraft sampled the early stages of this TS as it was still an AEW on 09 September (first panel, Fig. 2), and the resulting data are analyzed in our study.

AEW 5 moved off the west coast of Africa on 12 September, initially producing disorganized showers and thunderstorms, but eventually developing into TS Gaston on 20 September. Gaston dissipated by 25 September when additional strengthening was prevented by colder and drier air intrusion. Between 14 and 15 September, AEW 6 moved off the west coast, transforming

into TS Ian on 24 September and intensifying into a hurricane on 26 September. The research flight on 14 September flew between AEW 5 (which led to TS Gaston, third Panel Fig. 2) and AEW 6 (which led to Hurricane Ian, third Panel Fig. 2), while the flight on 15 September traversed north of AEW 6, sampling a dust outbreak located in the outer environment of the AEW (fourth Panel Fig. 2).

On 22 September, AEW 8 moved off the African coast, transforming into TS Hermine on 23 September before weakening

back to a tropical depression on 24 September. This TS which developed from AEW 8, coinciding with the highest concentrations of Saharan dust sampled during the CPEX-CV field campaign, was studied on 22 and 23 September (sixth and seventh panel, Fig. 2), and the resulting data are included in our analysis. Additional detailed information on developing AEWs is provided in Table 3.

### 3.2 Impact of aerosol on heating rates

Profiles of mean temperature in Kelvin (top row), specific humidity in kg/kg (middle row), and aerosol extinction coefficient in $km^{-1}$ (bottom row) which were utilized in the radiative transfer calculation are depicted in Fig. 3. The grey shading shows the spread of all profiles ingested in the RTM, for both background and anomalous aerosol concentrations. CPEX-CV data is shown in the left column (where temperature and humidity data are from the AVAPS dropsonde dataset, and extinction coefficient data from the HALO dataset), while the other two columns show the collocated mean profiles and corresponding spread

obtained from MERRA-2 and CAMS reanalysis. The temperature and moisture profiles across the three datasets exhibit striking similarities. However, the specific humidity profiles exhibit high variability across the three datasets (CPEX-CV, MERRA-2, and CAMS), as seen from the large spread of profiles specifically between the surface and 800 hPa. This variability impacts heating rate profiles, which is especially noticeable at these levels for heating rates calculated from the CAMS dataset.

On the other hand, comparisons reveal significant disparities when examining aerosol extinction coefficient profiles. At

this point, it is noteworthy to remind the reader that MERRA-2 assimilated data collected during CPEX-CV. Despite the assimilation of CPEX-CV data into MERRA-2, MERRA extinction coefficient profiles exhibit much higher surface extinction coefficient compared to the CPEX-CV data in the atmospheric layer between 1000 – 900 hPa). However, both MERRA-2 and CAMS miss most of the variability in extinction coefficient throughout the tropospheric column that the HALO profiles capture, including several notable aerosol layers around 800, 700, and 550 hPa. In the case of MERRA-2, most of the aerosol

is confined to the lower levels of the atmosphere. The contrast becomes even more pronounced when comparing the CPEX-CV and MERRA-2 against CAMS extinction coefficient. CAMS underestimates the extinction coefficient by an order of magnitude in several portions of the troposphere. Interestingly enough, the differences found for aerosol loading are not only confined to the extinction coefficient profiles, but there are also noticeable differences in AOD. The mean AOD for background and

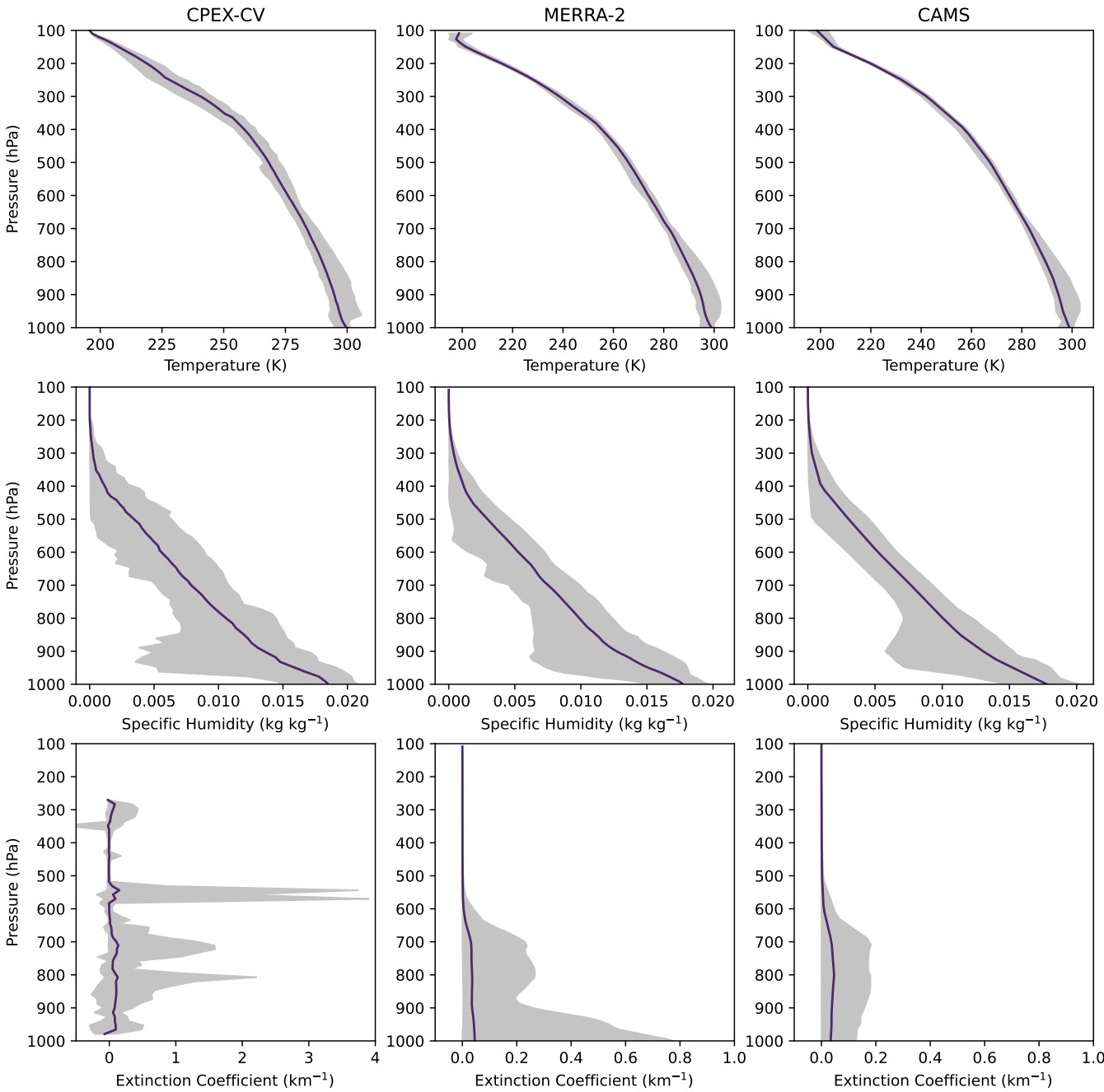

**Figure 3.** Temperature, specific humidity and extinction coefficient mean (purple) and range (grey shading) for all CPEX-CV, MERRA-2 and CAMS profiles used in the study, including background and anomalous dust concentrations (note the difference in the x-axis scale between the leftmost extinction coefficient panel and the center and rightmost panels).

anomalous cases over the seven research flights for CPEX-CV, MERRA-2, and CAMS are summarized in Table 2. There
are significant differences in the mean observed AODs and the ones provided by the reanalysis. In general, the reanalysis
overestimates the AOD compared to the observations for background cases and underestimates it for the anomalous cases.
These differences result in significant discrepancies in the heating rate calculations, as we will discuss below.

As mentioned above, 64 different cases were identified, all within a developing AEW or its environment. For each one of
these cases, we run the RTM for 3 datasets (CPEX-CV, MERRA-2, and CAMS), and for each of them, two RTM runs are
performed: one without the aerosol effect (RTM only initialized with pressure, temperature, specific humidity, and ozone),
which we refer to as the control run, and another one using the same atmospheric information with an added aerosol extinction
coefficient profile and corresponding AOD, which we refer to as the aerosol-aware run, for a total of 384 runs. Fluxes (in W/m$^2$)
and heating rates (in K/day) are thus calculated for each one of the 384 cases. To better understand the impact of different dust
loading scenarios, we categorize the data into two groups based on AOD as sampled during CPEX-CV. We thus define as
background all profiles with AOD $\leq$ 0.2 and select an equal number of profiles (32 profiles) with the highest AOD which we
define as "anomalous" or high (where the minimum AOD that meets this criteria is 0.335). This approach resulted in a 50-50
data split; in other words, 50% of the profiles are labeled as anomalous or high dust, while the lower 50% was classified as low
or background dust.

We proceed to subtract the control run from the aerosol-aware run at each profile location to evaluate the impacts of dust
on heating rates. We calculate the mean and corresponding standard deviations for our aerosol-aware minus control profiles.
Figure 4 shows the mean heating rate differences (aerosol-aware minus control) for background dust concentration and Fig.
5 shows the same for anomalous dust concentrations. Heating rate differentials are provided in three panels: shortwave (SW),
longwave/infrared (LW/IR), and total heating rate differences between the aerosol-aware run and the control run.

Figure 4 reveals notable discrepancies between datasets in the calculated heating rate differences, particularly in the SW
contribution. The differences in SW heating rate profiles, especially the significant divergence below 800 hPa, can be attributed
to variations in distribution (as indicated by extinction coefficient profiles), specifically the differences in AOD detailed in Table
2. CAMS reports the highest mean AOD for the background cases, and this is reflected in both the vertical distribution and
magnitude of heating rates, particularly below 900 hPa. Both reanalysis profiles exhibit significantly higher SW heating rates
near the surface, exceeding 1 K, while the CPEX-CV heating is no larger than 0.4 K at the surface. Most of the contribution
to total heating rates comes from SW processes rather than IR. The fact that most CPEX-CV sampling occurred during the
morning and close to solar noon may explain some of this behavior, but is also an artifact of the choice of optical properties
within the FLG RTM. Saharan dust, often composed of mineral-rich particles, is very active in the SW. Consequently, the
presence of Saharan dust in the atmosphere leads to the absorption of a significant portion of SW radiation, resulting in the
localized heating effects we observe in the heating rates differential.

The much smaller disparity observed between aerosol and control runs in the LW/IR radiation can be explained by the
inherent characteristics of the optical properties for transported dust within the model. Unlike certain aerosols such as sulfates
and nitrates that highly influence LW/IR radiation, mineral dust aerosols, including those from Saharan dust, tend to exhibit
lower absorption efficiency in the LW/IR spectrum. The contribution of LW/IR radiation to radiative forcing is further limited

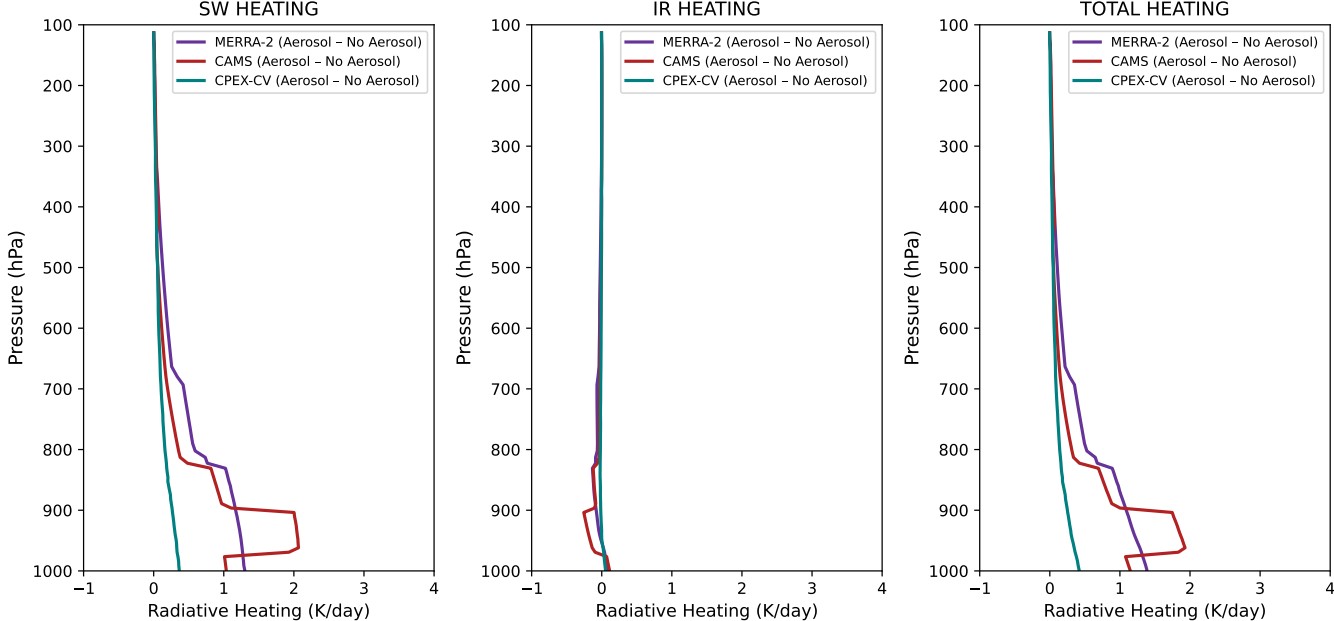

**Figure 4.** Heating rate difference between aerosol-aware and control run for MERRA-2 (purple), CAMS (red) and CPEX-CV (teal) for the background dust case. The left panel shows SW heating, the center panel shows LW/IR heating and the right panel shows total heating.

by the dominance of the scattering effects of SW radiation by dust. Additionally, the interaction of various radiative forcing
components, including water vapor and greenhouse gases, may overshadow the specific impact of dust aerosols in the LW/IR region.

Figure 5 displays the same calculation as Fig. 4 but for heating rates calculated for anomalous dust profiles with AOD exceeding 0.335 as defined previously. The impact of anomalous dust on heating rates is evident when compared with Fig. 4; higher AOD values correspond to higher heating rates. The MERRA-2 SW heating difference reaches up to 2.2 K/day, while
the CAMS SW heating rate difference reaches up to 3.8 K/day. Notable differences with Fig. 4 are observed in the reanalysis data, where heating rates are higher than those calculated from CPEX-CV data below 800 hPa but lower than those calculated from CPEX-CV data between 700 and 250 hPa. The differences between reanalysis and CPEX-CV heating rates are shown in Fig. 6 and 7 and discussed in the next section. There is also a notable increase in LW/IR cooling below 800 hPa in both MERRA-2 and CAMS in Fig. 5 in comparison with the findings in Fig. 4, and an increase in LW/IR heating in the surface
levels. However, the total heating remains driven by the SW heating.

### 3.3 Dataset comparison

The impact of assimilating CPEX-CV data into the MERRA-2 reanalysis was assessed by differencing MERRA-2 heating rates with the heating rates calculated from observational data from CPEX-CV. This difference in heating rate is contrasted

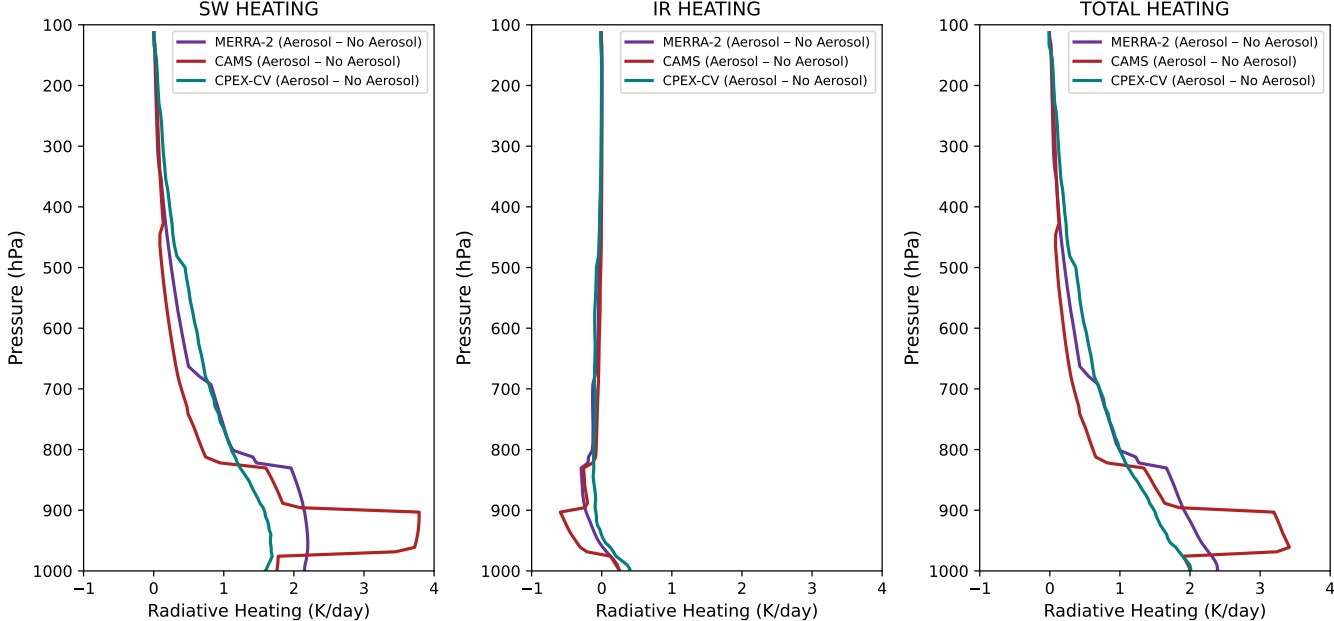

**Figure 5.** Heating rate difference between aerosol-aware and control run for MERRA-2 (purple), CAMS (red) and CPEX-CV (teal) for the anomalous dust case. The left panel shows SW heating, the center panel shows LW/IR heating and the right panel shows total heating.

with the difference between CAMS heating rates and observation (CPEX-CV), because CAMS did not assimilate CPEX-CV
data. This comparison of reanalysis minus observation sheds light on the performance of MERRA-2 in accurately represent-
ing aerosol-induced heating rates after the assimilation of high vertical resolution aerosol extinction coefficient profiles. The
differences between the reanalysis and observation were examined for both aerosol-aware runs and control runs, focusing on
both background and anomalous dust concentrations. In Fig. 6, the differences for background AOD (AOD < 0.2) are depicted,
while Fig. 7 illustrates the differences for anomalous AOD. In both figures, the purple line represents the heating rate difference
between MERRA-2 and CPEX-CV (MERRA-2 minus CPEX-CV), and the red line represents the heating rate difference be-
tween CAMS and CPEX-CV (CAMS minus CPEX-CV). The solid lines correspond to the aerosol-aware run, while the dotted
lines represent the control run.

   Examining the MERRA-2 background AOD case in Fig. 6, the difference in SW heating reaches 0.9 K/day at the surface
in the aerosol-aware run (solid purple line), and drops significantly around 825 hPa. The largest dust-induced SW heating
differences between MERRA-2 and CPEX-CV are thus at these lower levels of the atmosphere. Strong differences in LW/IR
heating are seen for MERRA-2, exceeding 1.55 K/day around 600 hPa. The control run LW/IR profiles are very similar
to the aerosol-aware profiles, highlighting the lack of LW/IR interaction of dust with LW/IR radiation. For anomalous dust
concentrations in Fig. 7, the aerosol-aware run exhibits a similar SW heating profile to the background dust profile in Fig.
6 for MERRA-2 below 825 hPa, with a maximum of 0.6 K/day at the surface. This suggests that MERRA-2 represents the

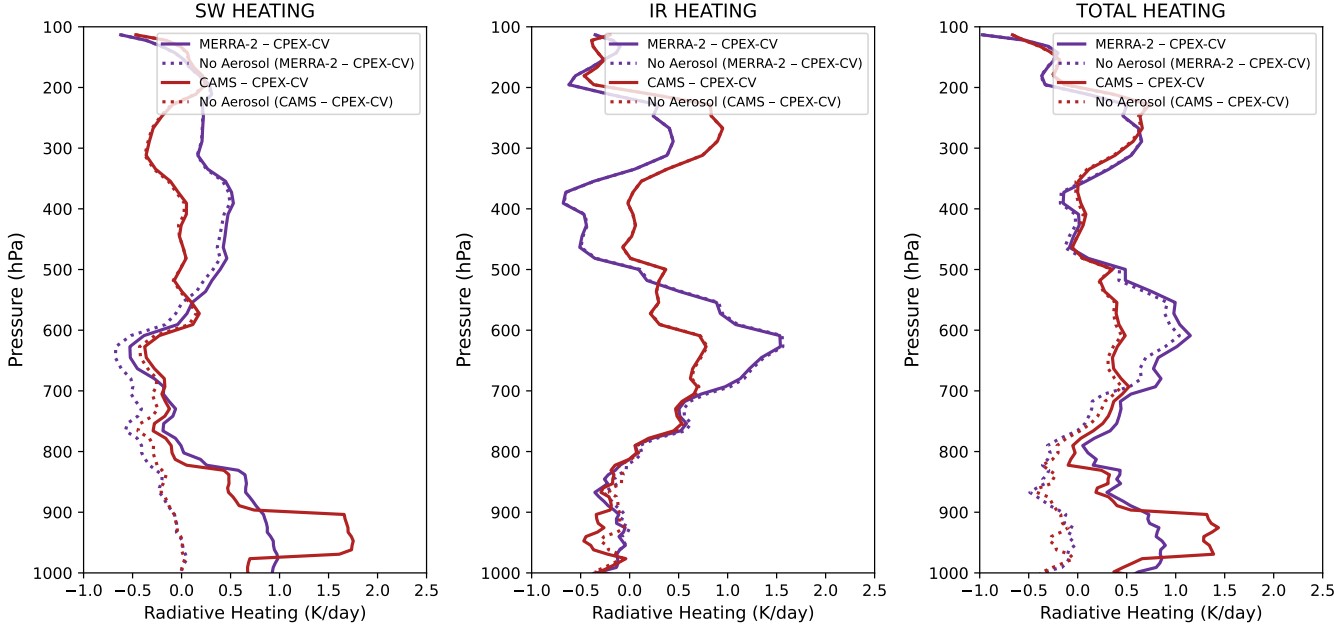

**Figure 6.** Heating rate difference between reanalysis and observation for the background dust case. Differences between MERRA-2 and CPEX-CV are plotted in purple and differences between CAMS and CPEX-CV are plotted in red. The solid lines correspond to the aerosol-aware run and the dotted lines correspond to the control run.

observation better at higher dust concentrations. The LW/IR heating profile exhibits similar differences to the background dust case depicted in Fig. 6, consistent with the fact that dust does not interact strongly with LW/IR radiation.

For both background AOD and anomalous AOD cases, as shown in Fig. 6 and 7, the SW heating difference between CAMS and observation is around 1.7 K/day between 975 hPa and 900 hPa in the aerosol-aware run for the background AOD case, and up to 2.05 K/day for anomalous dust cases. This discrepancy was found to be driven by the CAMS humidity profile at this atmospheric level. Large discrepancies in LW/IR between CAMS and observation are evident between 700 hPa and 500 hPa for both aerosol-aware and control runs, reaching up to 0.75 K/day for background AOD and 1.2 K/day for anomalous AOD cases. The LW/IR heating difference between CAMS and observation is smaller between 700 hPa and 550 hPa than those observed between MERRA-2 and observation, but is larger above 500 hPa. However, these differences are not likely driven by errors in dust aerosol characterization since there are minimal differences between the control run and the aerosol-aware run.

A major result from Fig. 6 and 7. is that despite the assimilation of CPEX-CV HALO aerosol profiles into the MERRA-2 reanalysis dataset, large differences in SW rates persist throughout the atmosphere between MERRA-2 and what was observed during CPEX-CV, and that these differences are driven by the aerosol profile. The average aerosol-aware MERRA-2 SW heating difference with CPEX-CV is 0.37 K/day, and the average aerosol-aware CAMS SW heating difference with CPEX-CV is 0.54 K/day. While MERRA-2 performs better than CAMS at representing observation, heating rate differences of the

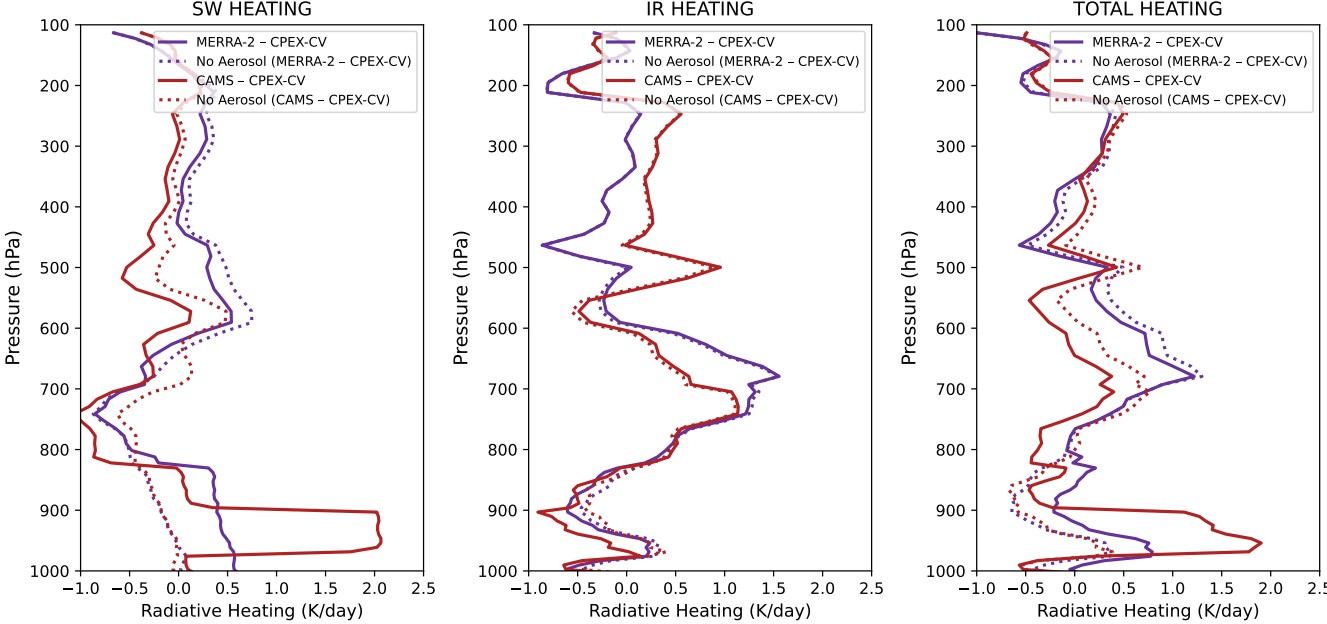

**Figure 7.** Heating rate difference between reanalysis and observation for the anomalous dust case. Differences between MERRA-2 and CPEX-CV are plotted in purple and differences between CAMS and CPEX-CV are plotted in red. The solid lines correspond to the aerosol-aware run and the dotted lines correspond to the control run.

magnitude shown in these figures have a non-negligible effect on the atmosphere and cannot be ignored in modeling without repercussions on outputs.

### 3.4 Comparison with an extreme dust event: The June 2020 Godzilla dust storm

To elucidate the distinctions in aerosol representation and subsequent impacts on heating rates between the MERRA-2 and CAMS reanalyses, specifically in the absence of cloud-related influences as encountered in the context of AEWs, we conducted a comparative analysis. This investigation focused on a notable event known as the Godzilla dust storm, an extreme dust storm that peaked on 18 June 2020 in a cloud-free environment within the same geographic region. Notably documented in the literature (Yu et al., 2021), the event showcased unprecedented AOD levels, as depicted in Fig. 8 for 18 June 2020. The FLG RTM was employed to compute heating rates for a profile situated at 15°N and 20°W at 12:00Z, with a corresponding AOD of 2.70. The resultant aerosol impact, as illustrated in Fig. 9, accentuates the increase in SW heating profiles. CAMS exhibits pronounced SW heating concentrated between 950 hPa and 900 hPa, while MERRA-2 displays lower peak values but a broader range extending from the surface to around 800 hPa. The LW/IR heating differences are a few degrees above 0 K/day at the surface levels and remain close to 0 K/day day at higher levels for both MERRA-2 and CAMS. The aerosol heating is thus mainly driven by SW radiation. The findings from this comparative analysis align with our analysis of heating rates during

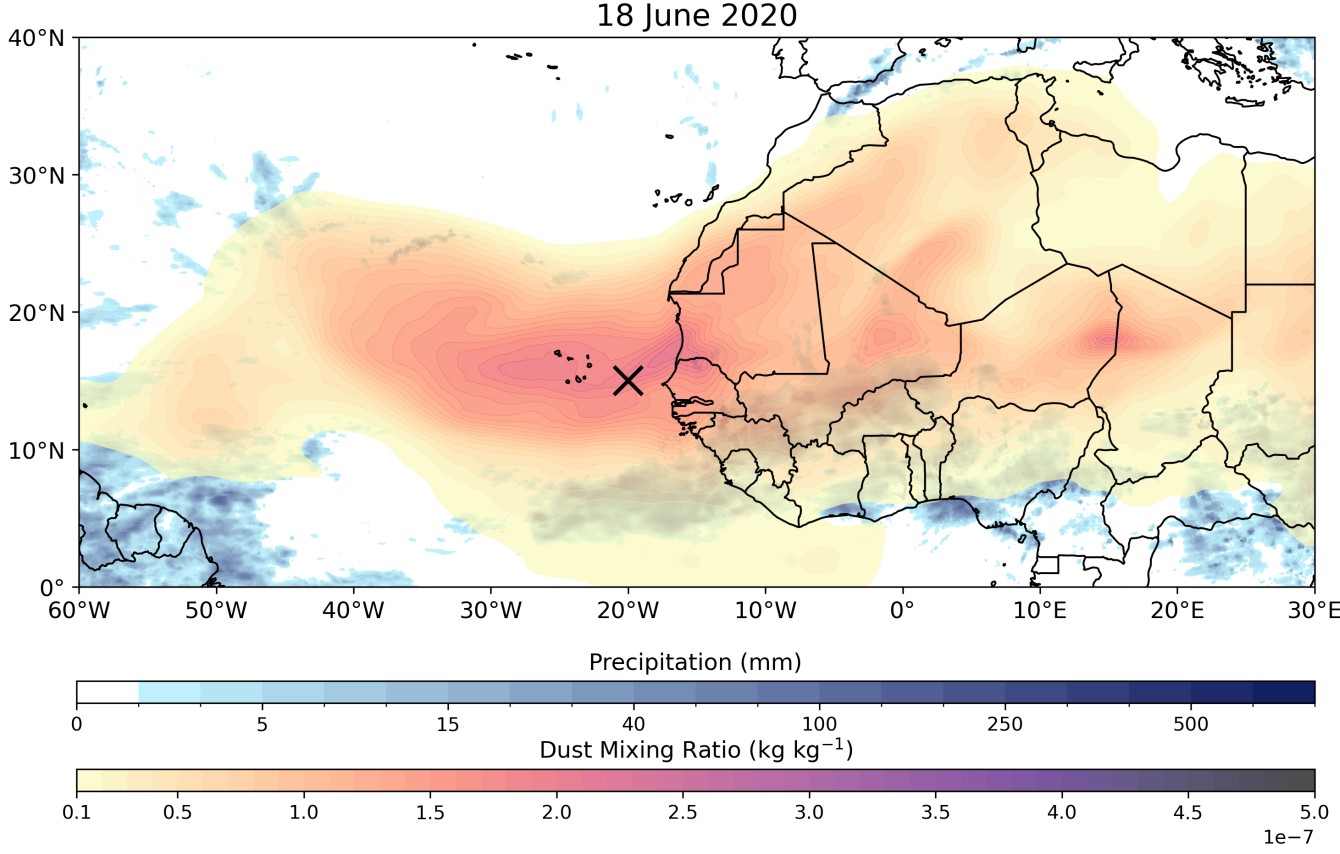

**Figure 8.** MERRA-2 total dust mixing ratio and IMERG daily accumulated precipitation on 18 June 2020. The location of the profile used in the analysis is marked in black.

the CPEX-CV campaign, reinforcing the robustness of the observed profiles and the utility of the FLG RTM in capturing the
nuances of aerosol-induced heating variations outside of cloud-influenced scenarios associated with AEWs.

### 3.5 Case study: Hurricane Fiona and TS Hermine

We analyze the effect of aerosol on heating rates using all three datasets on two days of interest (9 September 2022 and 22 September 2022) in the context of developing AEWs. The RTM simulations are performed after accounting for the solar zenith angle at the mean local time and location of the flight. The research flight on 09 September flew through AEW 4, which later
developed into Hurricane Fiona. The research flight on 22 September flew through AEW 8, which soon after developed into TS Hermine. Table 4 shows the mean, maximum, and standard deviation of AOD on both days studied as measured by the three datasets used in the case study, where the MERRA-2 and CAMS values are calculated from the collocation with the CPEX-CV flight path. The mean AOD was higher for the three datasets on 22 September 2022 than on 09 September, and this difference

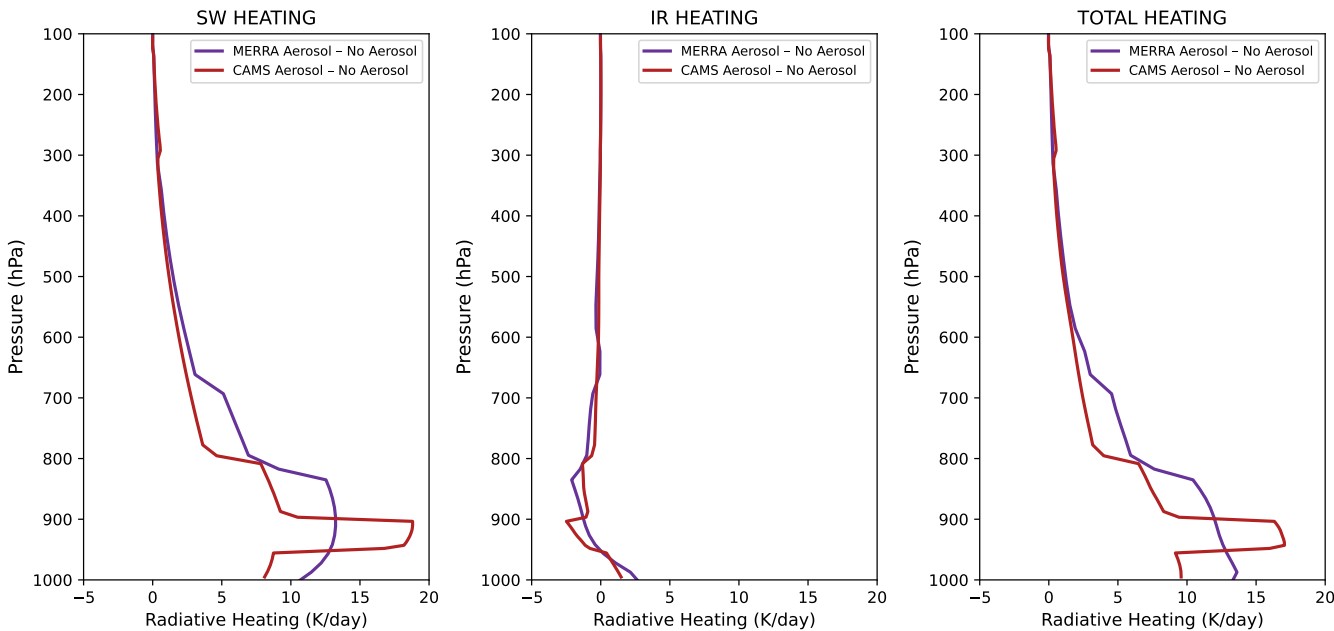

**Figure 9.** Heating rate difference between aerosol-aware and control run for MERRA-2 (purple) and CAMS (red) at 15°N, 20°W on 18 June 2020. The left panel shows SW heating, the center panel shows LW/IR heating and the right panel shows total heating.

**Table 4.** Mean, maximum and standard deviation of AOD for CPEX-CV, MERRA-2 and CAMS on 09 and 22 September 2022.

| Dataset | 09 September 2022 | | | 22 September 2022 | | |
|---------|---------|---------|---------|---------|---------|---------|
| | Mean AOD | Maximum AOD | Standard Deviation | Mean AOD | Maximum AOD | Standard Deviation |
| CPEX-CV | 0.25 | 1.69 | 0.16 | 1.02 | 3.34 | 0.64 |
| MERRA-2 | 0.33 | 0.72 | 0.16 | 0.59 | 2.49 | 0.48 |
| CAMS | 0.32 | 0.69 | 0.17 | 0.59 | 1.30 | 0.34 |

in the CPEX-CV dataset is nearly two times larger than the difference between the two days in the reanalysis datasets. Of note is also the maximum AOD of 3.34 captured by CPEX-CV, which is significantly larger than the maximum AOD in the MERRA-2 dataset (2.49) and the CAMS dataset (1.30).

Figure 10 illustrates the difference in aerosol mean heating rates between the aerosol-aware and control runs during the research flight on 09 September, where the average AOD value was 0.25 according to the CPEX-CV dataset. The greatest difference between CPEX-CV and reanalysis datasets are once again seen in the SW heating profile. Figure 10 shows a SW heating of 1 K/day at the surface which decreases with height for the CPEX-CV dataset. Consistent with our analysis above, this SW heating rate is greatly overestimated by both the MERRA-2 and CAMS reanalyses, with a heating rate of 2 K/day at

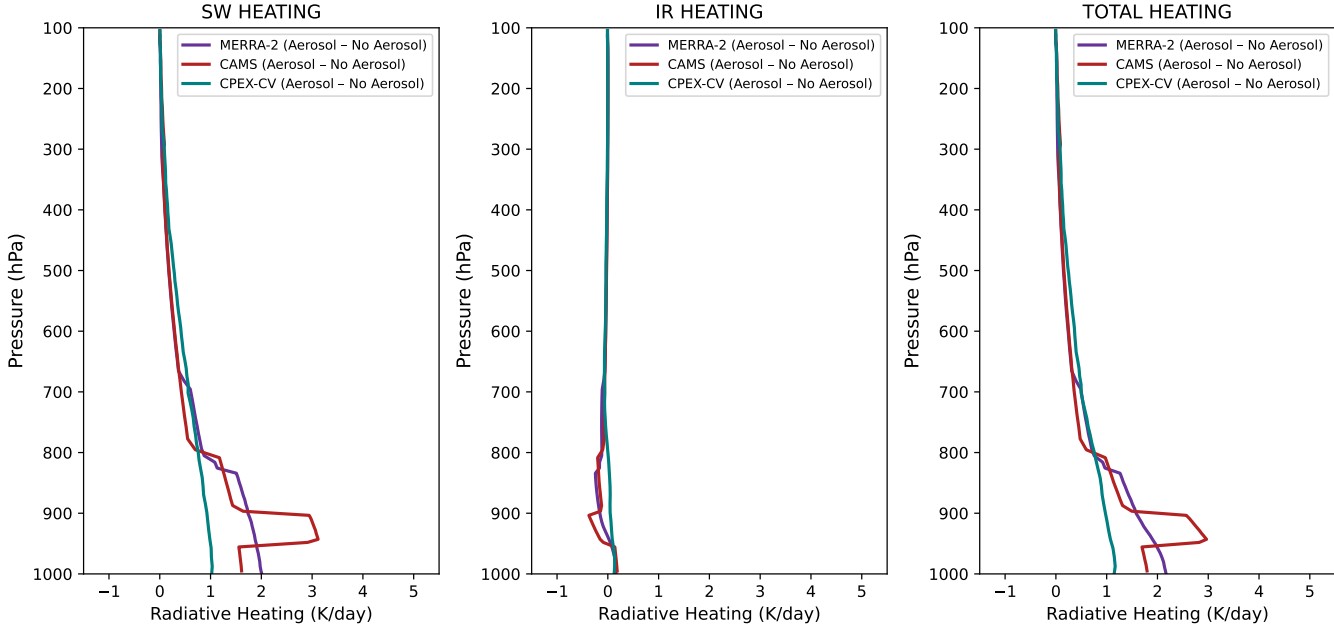

**Figure 10.** Heating rate difference between aerosol-aware and control run for MERRA-2 (purple), CAMS (red) and CPEX-CV (teal) on 09 September 2022. The left panel shows SW heating, the center panel shows LW/IR heating and the right panel shows total heating.

the surface for MERRA-2 and a heating rate of 1.6 K/day at the surface for CAMS, reaching up to 3 K/day between 950 hPa and 900 hPa.

Figure 11 shows striking differences in aerosol mean heating rates between the aerosol-aware and control run during the
research flight on 22 September, where the average AOD value was 1.02, over 4 times larger than on 09 September 2022. The CPEX-CV dataset shows a SW heating of 1.9 K/day at the surface, remaining between 1.9 K/day and 2.4 K/day up to 700 hPa, and decreasing with height above 800 hPa. This SW heating rate is once again overestimated by the MERRA-2 reanalysis below 800 hPa, with a heating rate of 2.3 K/day at the surface. The SW heating rate at the surface from CAMS is nearly identical to the heating rate from CPEX-CV, but the same increase in SW heating for CAMS between 950 hPa and 900 hPa attributed
to the humidity profile as seen in previous cases reaches 4.4 K/day. Above 800 hPa, the two reanalyses greatly underestimate the SW heating rate in this case, with differences of over 1 K/day between CPEX-CV and reanalysis. The CPEX-CV data reveals that the heating is evenly distributed throughout the column, whereas the reanalyses overestimate heating at the lower levels and underestimate heating at the upper levels. These differences in the structure of vertical heating will likely impact the forecast of the development of AEWs. High dust concentration has the potential to strongly alter the heating profile, as
illustrated in Fig. 11, exacerbating differences between reanalysis and observation. In the context of developing AEWs, the heating rate differences between observation (CPEX-CV) and reanalysis represent a significant inadequacy in atmospheric characterization, which must be addressed to avoid repercussions on model outputs of such systems. Furthermore, Table 4

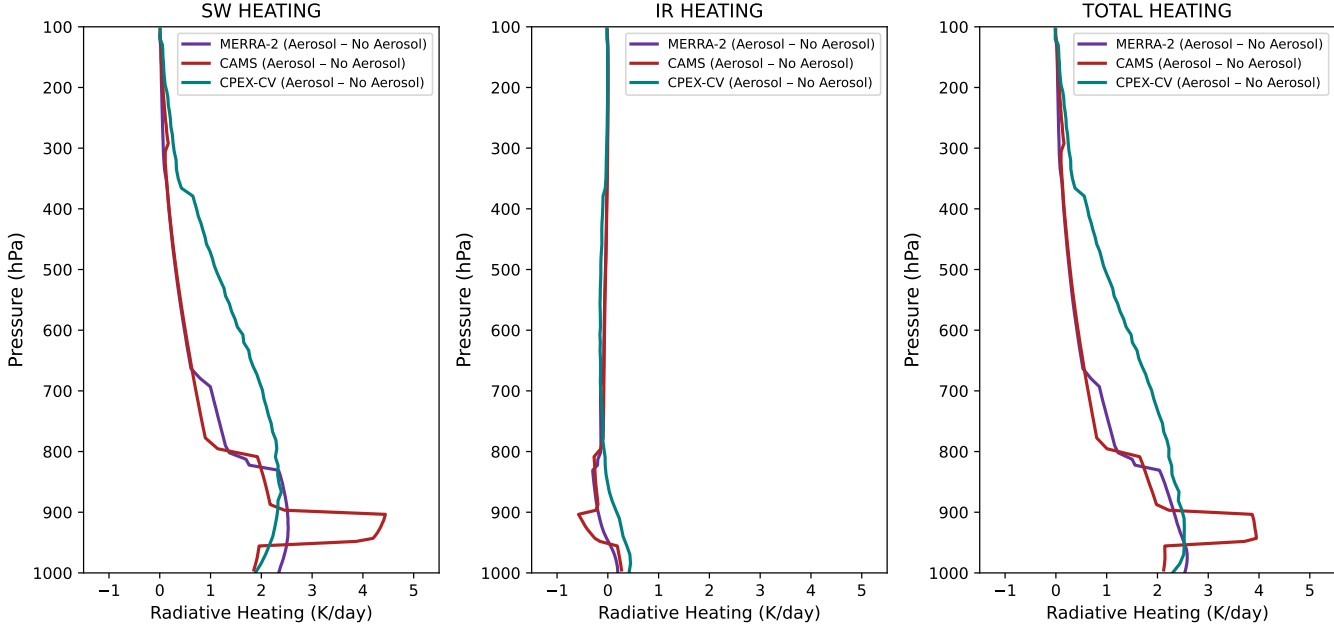

**Figure 11.** Heating rate difference between aerosol-aware and control run for MERRA-2 (purple), CAMS (red) and CPEX-CV (teal) on 22 September 2022. The left panel shows SW heating, the center panel shows LW/IR heating and the right panel shows total heating.

indicates the differences in AOD between CPEX-CV and reanalysis datasets are even greater at some locations, where CPEX-CV captured AOD values of up to 3.34, meaning that the differences captured in the mean heating rate profile in Fig. 11 may be even larger at specific profile locations.

We analyze the structure of the dust-induced shortwave heating throughout the progression of Fiona and Hermine. We use the AEW tracker documented in Lawton et al. (2022) to collocate MERRA-2 and CAMS profiles with the center of the AEW (and the TC it develops into), and calculate dust-induced (aerosol-aware minus control) SW heating rates. As shown previously, the dust-induced total heating rates are driven primarily by SW radiation. Because there is no SW activity during nighttime, only daytime profiles are studied. The profiles shown in Fig. 12 correspond to profiles at the closest gridpoint to the center of the AEW/TC as determined by the AEW tracker. The first profile corresponds to the first time-step where the center of the developing AEW was located over the ocean rather than land. The heating rate evolution is plotted for the development of the AEW into a TS. The last profile for Fiona corresponds to the last time-step before Fiona made landfall and subsequently weakened, whereas the last profile for Hermine corresponds to the last time-step before the TS became a post-tropical remnant low. Figure 12 reveals important differences between the dust-induced SW heating during Fiona's development in comparison to Hermine's. Based on the previous comparisons of heating rates calculated from reanalysis against those from observational data, we have determined that the MERRA-2 dataset captures the most accurate representation of the vertical structure of heating of the two reanalyses. In Fig. 12, MERRA-2 shows that the heating below 800 hPa varies between values of 0.52 K

and 1.59 K/day throughout the development of Fiona. On the other hand, heating at 12 UTC on 23 and 24 September 2022

reaches up to 2.78 K/day. This large difference in SW heating is noteworthy in the context of the short lifespan of Hermine, which was unable to intensify to the scale of Fiona (Category 4 Hurricane). The high degree of heating in the last profile for Fiona can be attributed to an anomalously high AOD value (0.64), while in previous time steps, AOD remained between 0.07 and 0.20. The max AOD value for the profiles plotted during Hermine was 0.4. The CAMS profiles show similar results to the MERRA-2 dataset, but still overestimate the SW heating around 900 hPa. The heating at 12 UTC on 24 September 2022 is

also much lower for CAMS with a peak heating value of 1.34 K/day versus 2.64 K/day for MERRA-2.

Over the ocean, Sun and Zhao (2020) finds that dust tends to reduce specific humidity in the lower troposphere, particularly in regions with high aerosol loading, while simultaneously augmenting midlevel moisture levels . They also find that dust warms the lower troposphere, promoting convection and generating positive vorticity between approximately 800–1,000 hPa, where most of the aforementioned SW-induced heating rates in both reanalysis prevail. This warming effect can enhance vertical

wind shear and consequently impacts environmental conditions in TC genesis regions. While we recognize this is not enough to draw a conclusion, since microphysics are not considered, and isolating the impact of thermodynamics to just aerosols is difficult, the large differences in heating at the lower levels of the atmosphere between the two cases raise the question of the impact of dust-induced radiative heating on AEW development.

### 3.6 A note on clouds

It is crucial to emphasize that our investigation, utilizing CPEX-CV data and the Godzilla dust storm event, provides valuable insights into aerosol-induced heating variations. Radiation calculations primarily rely on dust fields and include parameters such as pressure, temperature, moisture, and ozone profiles. The model acknowledges clouds based on moisture profiles but does not explicitly represent critical factors such as liquid water path and specific optical properties associated with clouds. This recognition highlights the limited scope of the cloud-related information provided by the model, underscoring the necessity

for future research to integrate a more comprehensive treatment of cloud-related variables for a detailed understanding of atmospheric interactions.

### 4 Conclusions

The paper explores the impact of Saharan dust plumes on atmospheric heating rates in the context of African Easterly Wave (AEW) development using radiative examination techniques based on reanalysis and NASA airborne observations. The study

leverages data from the Convective Processes Experiment – Cabo Verde (CPEX-CV) and multiple reanalysis datasets, including MERRA-2 and CAMS. The study examined data from seven DC-8 flights during the CPEX-CV field campaign, corresponding to 10 different AEWs, with a special emphasis on four waves that developed into named tropical storms, two intensifying into hurricanes. The primary objectives include assessing the accuracy of reanalysis in depicting aerosol radiative properties, comparing the impact of Saharan dust on atmospheric heating rates in different AEW scenarios, and evaluating the impact

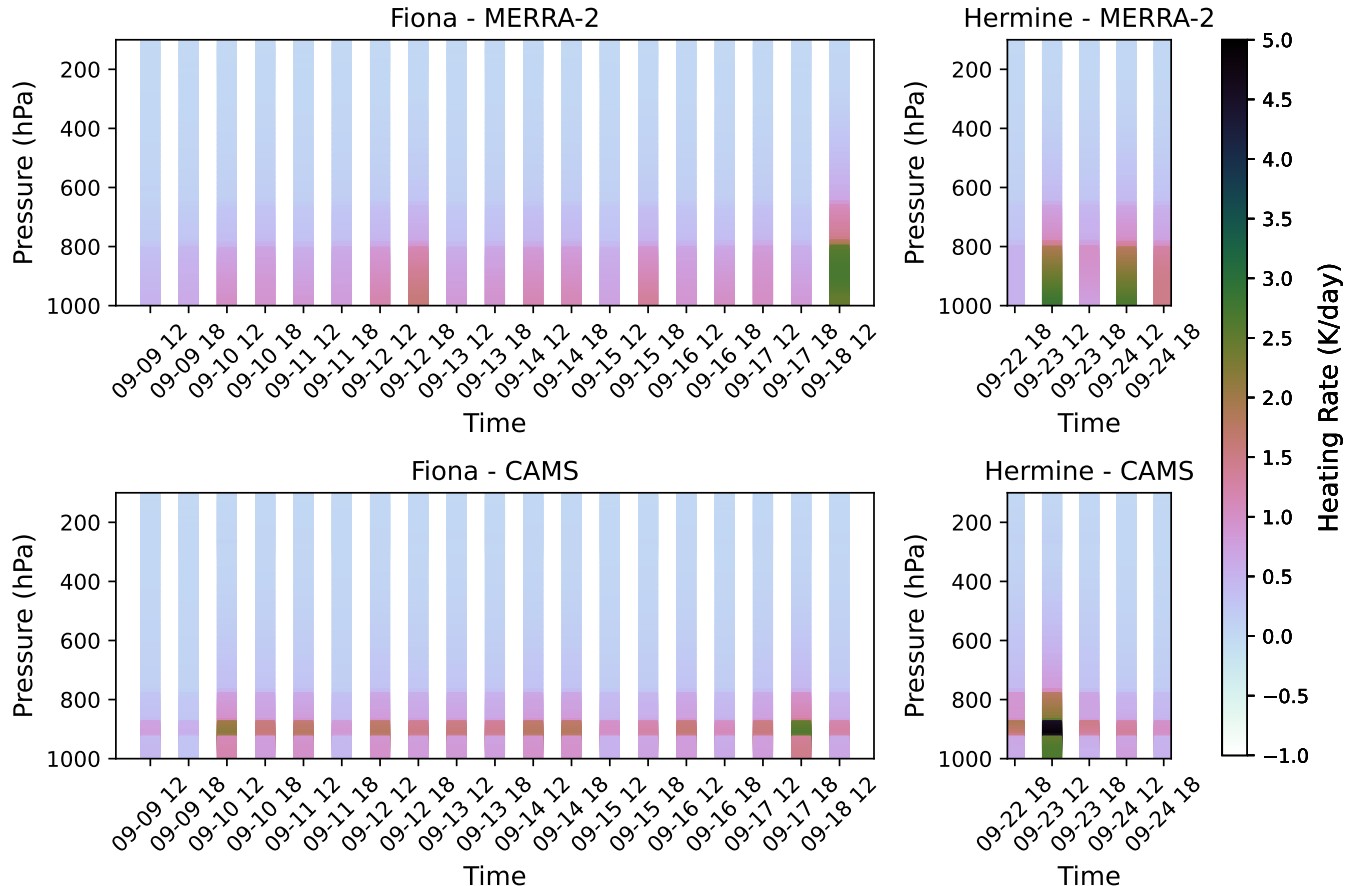

**Figure 12.** Vertical profiles of heating rates (K/day) at the center of two developing AEWs following their development into named tropical storms (Fiona, left and Hermine, right) for two reanalysis datasets (MERRA-2, top and CAMS, bottom).

of aerosol assimilation on model representation. The methodology integrates observational data from CPEX-CV with a four-stream radiative transfer model (Fu-Liou-Gu RTM), utilizing aerosol profiles from MERRA-2 and CAMS reanalyses.

The research revealed significant variations in aerosol-induced heating rates between observational data (CPEX-CV) and reanalyses (MERRA-2 and CAMS). The reanalyses exhibited significant differences in extinction and atmospheric profiles compared to observed data, impacting the calculation of total heating rates. Both MERRA-2 and CAMS radiative transfer runs consistently overestimated shortwave (SW) heating rates below 800 hPa, with errors up to 2.05 K for the CAMS dataset in anomalous dust cases, and errors up to 0.6 K for MERRA-2 even after assimilating CPEX-CV data. This was due to notable disparities in the representation of aerosol in extinction coefficient and aerosol optical depth (AOD). MERRA-2 exhibited higher surface extinction coefficient than what was measured by CPEX-CV at the surface but missed the variability throughout the tropospheric column that was captured by the observational data, revealing challenges in accurately representing aerosol

effects in models. Errors in CAMS humidity and extinction profiles resulted in an overestimation of heating below 900 hPa. Both reanalyses also exhibited too high AOD values in background cases as contrasted with CPEX-CV data and too low values in the anomalous cases. A comparative analysis of an extreme dust event (June 2020 Godzilla dust storm) reinforced the findings, showcasing differences of over 5K in SW heating profiles between MERRA-2 and CAMS. This analysis provided further evidence of the robustness of observed profiles and the model's ability to capture aerosol-induced heating variations. Finally, a case study focusing on Hurricane Fiona and Tropical Storm Hermine illustrated the impact of aerosols on heating rates during specific research flights. Both reanalyses exhibited notable discrepancies in SW heating rates compared to observed data, with potential implications for forecasting the development of AEWs.

When considering the impact of heating rates on TC development, a noteworthy observation emerges regarding the impact of varying dust concentrations on the process, particularly in relation to heating rates. The difference in the vertical heating profile between lower (AOD = 0.25) and higher (AOD = 1.02) dust concentrations as revealed on 09 and 22 September 2022 highlights the potential existence of a dust concentration threshold over which dust-induced atmospheric heating acts to affect the development of the system. Dust concentration and thus dust-induced radiative heating during the sampling of Pre-Fiona, which intensified to a Category 4 hurricane, was significantly lower than during the sampling of Hermine. Strikingly, when TS Hermine was sampled, characterized by elevated dust concentrations and higher dust-induced heating rates, the TS exhibited a subsequent weakening and a short-lived time span. Following the progression heating rate profiles throughout the development of the AEW, we notice significantly lower heating rates (0.52 K/day to 1.59 K/day) during Fiona's development than during Hermine (up to 2.78 K/day), raising the question of the role of dust induced heating in the development of these storms. Of course, it is essential to acknowledge that this analysis only addresses a finite part of a complex system, and a more comprehensive examination of environmental factors is imperative for a nuanced understanding. Despite this limitation, the observed patterns underscore the significance of incorporating dust-related variables in hurricane modeling studies. These results warrant a more in-depth investigation to elucidate the intricate interplay between dust concentrations, heating rates, and their influence on hurricane development, emphasizing the necessity for rigorous modeling studies to advance our comprehension of these intricate atmospheric phenomena. Furthermore, the vertical structure of the heating was inaccurately represented by the reanalyses, specifically in the case of Hermine, where heating was overestimated below 800 hPa but underestimated above 800 hPa, the CPEX-CV dataset revealing a much more uniform heating distribution than the two reanalyses. Such errors in the heating profile are likely to impact the modeling of the AEW development, and should be analyzed in future studies.

The research significantly advances our understanding of the importance of accurately characterizing aerosol-induced heating rates during AEW development. The findings underscore the limitations of current reanalysis datasets in accurately capturing aerosol properties and their radiative effects, particularly at critical atmospheric levels (1000-500 hPa). Despite the assimilation of observational data, substantial differences persist, revealing the need for further refinement in modeling aerosol dynamics. The study emphasizes the importance of considering vertical distribution and composition of aerosols in assessing their impact on AEWs. Unveiled by advanced radiative transfer modeling, the observed discrepancies in heating rates between reanalysis and airborne observations at key atmospheric levels have implications for weather forecasting, emphasizing the need

for improved aerosol parameterizations in NWP models, and provide valuable insights into the challenges and opportunities

for refining our understanding of aerosol-AEW interactions in the Atlantic basin.

*Data availability.*  The CPEX-CV data used in this study can be obtained from https://www-air.larc.nasa.gov/missions/cpex-cv/index.html (CPEX-CV, 2022). The MERRA-2 reanalysis data and surface parameters are available via NASA's Global Modeling and Assimilation Office website: https://gmao.gsfc.nasa.gov/reanalysis/MERRA-2/ and or via the NASA Goddard Earth Sciences (GES) Data and Information Services Center (DISC). The CAMS reanalysis can be found in https://atmosphere.copernicus.eu/data. Likewise, the Fu Liou Gu Radiative

Transfer model code is available via the UCLA (http://people.atmos.ucla.edu/gu/Fu-Liou-Gu_Radiative_Transfer_Model.htm).

*Author contributions.*  RWB led the radiative transfer study. RWB and MIOM designed the experiments. RWB was responsible for obtaining datasets used as input for the radiative transfer calculations. RWB conducted the radiative transfer modeling and performed the majority of the analysis and was supervised and assisted by MIOM. RWB and MIOM prepared the manuscript and figures.

*Competing interests.*  The authors declare that they have no conflict of interest.

*Acknowledgements.*  Authors Ruby W. Burgess and Mayra I. Oyola-Merced acknowledge the CPEX-CV team for collecting the data and making it available to the community (https://www-air.larc.nasa.gov/cgi-bin/ArcView/cpex.2022), as well as Professors Angela Rowe, Tristan L'Ecuyer, and Daniel Vimont for their feedback in the research process. This work was performed at the University of Wisconsin-Madison. Support for this research was provided by the Office of the Vice Chancellor for Research and Graduate Education at the University of Wisconsin–Madison with funding from the Wisconsin Alumni Research Foundation.

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
