# Peer review of "Saharan Dust Impact on Radiative Heating Rate Errors Inherent in Reanalysis Data in the African Easterly Wave Development Region"

_EGUsphere, 2023_

## Referee Comment (RC1)

ACP review of:

Radiative Examination of Developing African Easterly Waves and Saharan Dust Interactions: Comparative Insights from Reanalysis and NASA Airborne Observations

By Ruby W. Burgess and Mayra I. Oyola-Merced

General Comments:

This paper is reveals important information regarding the limitations of using reanalysis data to characterize the interactions between African Easterly Waves (AEWs) and Saharan dust, but the paper states that "Our primary objective is to perform a radiative examination of the interactions between AEWs and Saharan Dust during the intensive observation period (IOP)."  A radiative examination was performed for dust aerosol profiles in terms of their heating rates, but a radiative examination between AEWs and Saharan dust was not done since this would require an evaluation of the impact of the dust on clouds, and as stated on p. 19, this was not done.  That is, cloud-aerosol interactions were not studied. The title of this paper should be changed to reflect the actual content of the paper; something like "Radiative Heating Rate Errors Inherent in Reanalysis Data Used to Evaluate Saharan Dust Associated with African Easterly Waves".  And the objective of the paper could be restated accordingly; something like "Our primary objective is to perform an examination of radiative heating rates within Saharan dust plumes associated with AEWs during the intensive observation period (IOP)."  The scope of the research when mentioned elsewhere in the paper should be changed accordingly.

Other than this, the paper is well written and well organized, and the heating rate errors associated with the reanalysis data represent an important discovery.  I recommend publication pending minor revisions as indicated above and below.

Major comments:

1.  Line 102:  Lidar systems measure the aerosol backscatter, and from backscatter infer the extinction coefficient.  So, a reference indicating how the 532 nm aerosol extinction coefficient was calculated from the backscatter would be prudent here.

2.  Lines 115-116: Suggest using "volume extinction coefficient" rather than mere "extinction coefficient" to avoid confusion with the mass extinction coefficient used in Eq. 1.

3.  Equation 3:  It would be instructive to briefly note that the first two terms tend to dominate the heating rate magnitude, with the first term in brackets predicting radiative exchange with the boundary layer (and thus is generally a heating term) while the second term predicts radiative exchange with the top of atmosphere and thus for longwave radiation predicts cooling to space.

4.  Lines 386-389:  This sentence states: "The primary objectives include assessing the accuracy of reanalysis in depicting aerosol radiative properties, **understanding the influence of aerosol heating rates, especially from Saharan dust, on AEW development**, and evaluating the impact of aerosol assimilation on model representation."  The section in bold font does not appear

correct since it was not shown how heating rate errors involving Saharan dust affect the development of AEWs.

5. Lines 424-426:  This last sentence under "Conclusions" is an overstatement (similar to comment #4 above).  For example, the approach taken was not "comprehensive" since the potential impact of dust aerosol-cloud interactions on AEW development was not accounted for.

Technical comments:

1. Lines 116-117:  "each of the five bins" => "each of the five size-bins"?

2. Line 242:  Two incidences where "second panel Fig. 2" => "third panel Fig. 2"?

3. Line 248:  "fourth and fifth panel, Fig. 2" => "sixth and seventh panel, Fig. 2"?

4. Line 264:  "650 hPa" => "550 hPa"?

5. Lines 365-367:  The last part of this sentence appears to be missing.

---

## Author Response (AR1)

[Author responses to comments are in italics.]

RC1:

General Comments:

R1: This paper is reveals important information regarding the limitations of using reanalysis data to characterize the interactions between African Easterly Waves (AEWs) and Saharan dust, but the paper states that "Our primary objective is to perform a radiative examination of the interactions between AEWs and Saharan Dust during the intensive observation period (IOP)." A radiative examination was performed for dust aerosol profiles in terms of their heating rates, but a radiative examination between AEWs and Saharan dust was not done since this would require an evaluation of the impact of the dust on clouds, and as stated on p. 19, this was not done. That is, cloud-aerosol interactions were not studied. The title of this paper should be changed to reflect the actual content of the paper; something like "Radiative Heating Rate Errors Inherent in Reanalysis Data Used to Evaluate Saharan Dust Associated with African Easterly Waves". And the objective of the paper could be restated accordingly; something like "Our primary objective is to perform an examination of radiative heating rates within Saharan dust plumes associated with AEWs during the intensive observation period (IOP)." The scope of the research when mentioned elsewhere in the paper should be changed accordingly.

*We appreciate Reviewer #1's suggestion to redefine the scope of the paper and update the title accordingly. The title was changed to "Saharan Dust Impact on Radiative Heating Rate Errors Inherent in Reanalysis Data in African Easterly Wave Development Region," in keeping with the reviewer's suggestion while emphasizing the impact of Saharan dust on the radiative heating errors, which is shown in the experiments comparing background dust concentration and anomalous dust concentration.*

*The text in the introduction and throughout the paper was also updated to reflect that although the effect of heating rate differences was not shown, the effect of Saharan dust on heating rates during different AEW scenarios was analyzed. The goal of the study may not have been clearly stated throughout the paper. The goal of this study is to shed light on the importance of accounting for the vertical distribution of Saharan dust in the context of AEW development. This is done by showing the impact of anomalous dust loadings on modifying atmospheric heating rates at critical atmospheric levels during AEW development. We show that these impacts are not well captured by reanalysis, which is important to consider in the context of improving the modeling of AEW development. The introduction, conclusion, and other sections listed below were updated to clarify these goals.*

R1: Other than this, the paper is well written and well organized, and the heating rate errors associated with the reanalysis data represent an important discovery. I recommend publication pending minor revisions as indicated above and below.

Major comments:

1. Line 102: Lidar systems measure the aerosol backscatter, and from backscatter infer the extinction coefficient.  So, a reference indicating how the 532 nm aerosol extinction coefficient was calculated from the backscatter would be prudent here.

*References to the calculation of extinction coefficient from the backscatter were added in section 2.1.2.*

2. Lines 115-116: Suggest using "volume extinction coefficient" rather than mere "extinction coefficient" to avoid confusion with the mass extinction coefficient used in Eq. 1.

*The term "extinction coefficient" was changed to "volume extinction coefficient" in line 115-116 to avoid confusion.*

3. Equation 3: It would be instructive to briefly note that the first two terms tend to dominate the heating rate magnitude, with the first term in brackets predicting radiative exchange with the boundary layer (and thus is generally a heating term) while the second term predicts radiative exchange with the top of atmosphere and thus for longwave radiation predicts cooling to space.

*These suggestions were taken into consideration and added to the text in section 2.6.2.*

4. Lines 386-389: This sentence states: "The primary objectives include assessing the accuracy of reanalysis in depicting aerosol radiative properties, understanding the influence of aerosol heating rates, especially from Saharan dust, on AEW development, and evaluating the impact of aerosol assimilation on model representation."  The section in bold font does not appear correct since it was not shown how heating rate errors involving Saharan dust affect the development of AEWs.

*The text in the conclusion was updated to reflect that although the effect of heating rate differences was not shown, the effect of Saharan dust on heating rates during different AEW scenarios was analyzed. Lines 386-389 (now 464-466) were changed to "The primary objectives include assessing the accuracy of reanalysis in depicting aerosol radiative properties, comparing the impact of Saharan dust on atmospheric heating rates in different AEW scenarios, and evaluating the impact of aerosol assimilation on model representation."*

5. Lines 424-426: This last sentence under "Conclusions" is an overstatement (similar to comment #4 above).  For example, the approach taken was not "comprehensive" since the potential impact of dust aerosol-cloud interactions on AEW development was not accounted for.

*This sentence was removed, and the last paragraph of the conclusion was rephrased to account for this suggestion, along the lines of the response to comment #4.*

Technical comments:

1. Lines 116-117: "each of the five bins" => "each of the five size-bins"?

*The term "bins" was changed to "size-bins" in lines 117 and throughout the rest of the text.*

2. Line 242: Two incidences where "second panel Fig. 2" => "third panel Fig. 2"?

*"second panel Fig. 2" was changed to "third panel Fig. 2" in line 242, "third panel Fig. 2" was changed to "fourth panel Fig. 2" in line 244*

3. Line 248: "fourth and fifth panel, Fig. 2" => "sixth and seventh panel, Fig. 2"?

*"fourth and fifth panel, Fig. 2" was changed to "sixth and seventh panel, Fig. 2" in line 248.*

4. Line 264: "650 hPa" => "550 hPa"?

*"650 hPa" was changed to "550 hPa" in line 264.*

5. Line 365-367:  The last part of this sentence appears to be missing.

*The end of the sentence was removed to correct an editing error in the text.*

RC2:

General comments:

R2: The manuscript starts with a very nice well-written introduction section, but it has very little to do with the work presented in the rest of the paper, since the work doesn't address the issues raised in the introduction. Like the introduction, parts of the conclusion section have little relationship to the work presented.  For example,  "the paper explores the intricate interactions between African Easterly Waves (AEWs) and Saharan dust" is not talking about this paper.

*We appreciate the reviewer's feedback on this issue. The goal of this study is to shed light on the importance of accounting for the vertical distribution of Saharan dust in the context of AEW development. This is done by showing the impact of anomalous dust loadings on modifying atmospheric heating rates at critical atmospheric levels during AEW development. We show that these impacts are not well captured by reanalysis, which is important to consider in the context of improving the modeling of AEW development. The introduction, conclusion and other sections listed below were updated to make these goals more clear. The sentence "The paper explores the intricate interactions between African Easterly Waves (AEWs) and Saharan dust…" was changed to "The paper explores the impact of Saharan dust plumes on atmospheric heating rates in the context of African Easterly Wave (AEW) development using radiative examination techniques based on reanalysis and NASA airborne observations" in lines 459-460.*

A "primary objective" is "understanding the influence of aerosol heating rates, especially from Saharan dust, on AEW development", but this primary objective wasn't addressed.

*This line was rephrased to reflect that although the effect of heating rate differences was not shown, the effect of Saharan dust on heating rates during different AEW scenarios was analyzed. The wording "understanding the influence of aerosol heating rates, especially from Saharan dust, on AEW development" was changed to "comparing the impact of Saharan dust on atmospheric heating rates in different AEW scenarios" in lines 464-466.*

And "The distinction between anomalous and background dust cases introduces a fascinating dynamic, unveiling a correlation between lower dust levels during the AEW development and the formation of exceptionally potent hurricanes".  The authors may feel that they learned these things from the work they did examining the data, but they did not communicate these findings in a meaningful way to (at least one of) their readers. While I saw some dust maps and a list of which flights sampled airmasses that later became hurricanes, I did not see any discussion of a correlation or influence.

*We appreciate Reviewer #2's feedback on the phrasing of this section. We removed the reference to a correlation. The main point of this paragraph is to highlight that dust concentration (including in the vertical) is an important factor in terms of dust-induced heating and needs to be considered when studying AEW development. The text was reworded to emphasize these points, and the statement "The distinction between anomalous and background dust cases introduces a fascinating dynamic, unveiling a correlation between lower dust levels during the AEW development and the formation of exceptionally potent hurricanes" was changed to "The difference in the vertical heating profile between lower (AOD = 0.25) and higher (AOD = 1.02) dust concentrations as revealed on 09 and 22 September 2022 highlights the potential existence of a dust concentration threshold over which dust-induced atmospheric heating acts to affect the development of the system" in lines 485-488.*

*An additional figure (Fig. 12) was added to the manuscript to illustrate the evolution of heating rates throughout the development of the two AEWs studied and display the differences in the heating profile between the two. A paragraph was added to the methods section to describe the methodology used in this analysis: "We use the AEW tracker described in Lawton et al. (2022) to track the center of several AEWs of interest. The tracker calculates curvature vorticity at 700 hPa using the nondivergent component of the 700-hPa wind averaged within a radius of 600 km of each grid point. We use the positional dataset which supplies the location of the center of the storm at a 6-hour time step to collocate the center of the storm with the nearest MERRA-2 and CAMS reanalysis datasets." As stated in the results section, this figure "reveals important differences between the dust-induced SW heating during Fiona's development in comparison to Hermine's. The MERRA-2 dataset, which we have determined captures the most accurate representation of the vertical structure of heating of the two reanalyses, shows that the heating below 800 hPa varies between values of 0.52 K and 1.59 K/day throughout the development of Fiona. On the other hand, heating at 12 UTC on 23 and 24 September 2022 reaches up to 2.78 K/day. This large difference in SW heating is noteworthy in the context of the short lifespan of Hermine, which was unable to intensify to the scale of Fiona (Category 4 Hurricane). [...] The CAMS profiles show similar results to the MERRA-2 dataset, but still overestimate the SW*

*heating around 900 hPa. The heating at 12 UTC on 24 September 2022 is also much lower for CAMS with a peak heating value of 1.34 K/day versus 2.64 K/day for MERRA-2."*

R2: Within the main body, the authors have done a number of variations on the heating rate calculations, but any lessons learned from them are not clearly explained. For instance, the cases were stratified into subsets by AOT, but why? What was learned from it? There are case studies, but these are not analyzed in more detail than the bulk statistics.

*We show that Saharan dust, when in anomalous concentrations, has a non-negligible effect on atmospheric heating rates. Cases were not stratified into sets of AOD - we specify that AODs less than 0.2 are relatively small and normally considered "background" so all calculations were done from the standpoint that any case where the aerosol loading is of relevance is above 0.2. This allows us to show that above "background" concentrations, dust-induced heating significantly alters the heating rate profile, and thus is a key component to understanding atmospheric conditions during Saharan dust-affected AEW development. For anomalous AOD, dust impact on heating rates is non-negligible. The text was updated to make this point more clear. This result is important for the case study of Pre-Fiona and Hermine (Figs. 10-12), where AOD values bear significant differences between the two cases and have an important impact on observed heating rates. A table (Table 4) with data on mean, maximum and standard deviation of AOD values for each storm was also added and discussed to give more context to the reader on how these values differed between datasets and case study days.*

*Additional note: Subsection 2.6.3 was renamed "Heating rate experiments" to emphasize its role in the broader context of the section.*

R2: I would even say that it's not very clear what advantage there is to doing the heating rate calculations (instead of just stopping at the extinction comparison). The strongest conclusion, that the models represent the aerosol distributions poorly, could be illustrated by looking at extinction alone.

*It seems reviewer 2 may be missing the broader significance of calculating heating rates. While the primary focus of our study is indeed to assess aerosol distributions in models, the calculation of heating rates provides crucial insights beyond this scope. Aerosols interact with solar radiation in complex ways, absorbing or scattering sunlight and consequently influencing the atmospheric temperature profile. Understanding the spatial and temporal variations in heating rates due to aerosols is essential for accurately predicting atmospheric dynamics and weather phenomena. It's important to note that heating rates do not necessarily respond to increments in aerosol concentration in a linear way. (An easy way to do this is by examining the aerosol profile alongside the location of peak heating rates and see that they don't always correspond). This demonstrates the complexity of aerosol-atmosphere interactions and highlights the need for comprehensive assessments that go beyond simple extinction comparisons. For example, variations in temperature induced by aerosol heating can affect atmospheric stability, leading to changes in convection patterns and the formation of clouds and precipitation. Additionally, temperature inversions and boundary layer evolution, which play critical roles in the development of weather*

*systems such as African Easterly Waves (AEWs), are directly influenced by aerosol-induced heating. Furthermore, the humidity profile also impacts the heating rate profile, which is especially noticeable in the CAMS dataset below 900 hPa (mentioned in lines 280-281 of updated manuscript). Therefore, by including heating rate calculations, we gain deeper insights into the intricate interactions between aerosols and atmospheric dynamics, enabling more comprehensive assessments of model performance and improved predictions of weather and climate phenomena.*

R2: Unfortunately, further conclusions about heating rates are hampered by the fact that there is no observational truth for the heating rates, only heating rates calculated with all the same assumptions on observed extinction profiles.

*We invite reviewer 2 to re-read Section 2, where we provide comprehensive details about the radiative transfer model (FLG) utilized in our study, from which heating rates are obtained. This model is highly accurate and widely respected within the atmospheric radiation community for its reliability and accuracy. It's important to note that the output of the radiative transfer model is reliant on all observed parameters, some come from reanalysis and others from direct observations (CPEX-CV). Therefore, the heating rates derived from these observations represent the closest approximation to the "truth" available to us.*

R2: This still leaves open plenty of room for doubt about, e.g. whether the OPAC model for dust represents these cases accurately, the impact of the simplification of clouds in the radiative transfer calculation, etc.) (The heating rate calculations may nevertheless be valuable, though, to estimate the radiative impact of the errors in aerosol representation, although subject to potential errors in assumptions, which need to be discussed.)

*While these are valid points, the primary goal of our paper was not to evaluate the performance of radiative transfer model (FLG, which has been validated in many papers over the last few decades) nor the OPAC climatology. Yes, there are limitations, but circumventing them will require a lot of funding and effort from the community as a whole. FLG has been a staple in our field for decades and is widely respected for its accuracy and reliability. Similarly, while the OPAC climatology may require improvements, it remains the aerosol climatology of choice for many scientific organizations and Numerical Weather Prediction (NWP) centers.*

Line by line comments:

R2:  1. What instruments are the temperature and humidity data from in Figure 2 and in the CPEX-CV heating rate calculations?  In some places the manuscript suggests these are from the sondes (i.e. L96, L163) , but in others it suggests they are from HALO (i.e. Line 103, Line 168, Line 354. It needs to be more specific which instruments provide observational data, rather than just lumping all observational data under the label "CPEX-CV".  HALO does not measure temperature, and HALO temperatures provided in the HALO data files are from another source (often MERRA-2) and are provided for consistency since they are used in some of the HALO retrievals.  If the authors are using MERRA-2 temperatures "from HALO" and comparing them to

MERRA-2 that is obviously an error. I believe the HALO readme files explain this and that the source of the temperature and humidity data in the files is encoded in some way in the files.

*The temperature and humidity data labeled CPEX-CV in Fig. 3 are from the AVAPS dropsonde data files. We appreciate the reviewer's suggestion not to use the temperature and humidity profiles included in the HALO data files. We have modified the profiles in Figs. 10 and 11 to use a mean temperature and humidity from the dropsondes launched on each respective day (09 September 2022 and 22 September 2022) and adjusted the text accordingly, although only minor changes resulted. All figures pertaining to the CPEX-CV field campaign now only make use of temperature and humidity data from the AVAPS dropsonde data files.*

While on this subject, it is usual for users of field mission data to contact the measurement data providers to make sure the data is not misunderstood or misinterpreted (and a request to that effect is a standard part of field mission data readme files). I don't see any coauthors from the dropsonde team or the HALO team or acknowledgement of any disucssions with them. I suggest reaching out to these teams to get some backup support to prevent making errors with the interpretations of the datasets.

*From Co-Author Oyola-Merced: We understand and appreciate your suggestion regarding the involvement of a member of the dropsonde and HALO teams in the publication process. However, we disagree with the assertion that our publication should include a member of these teams as coauthors. While we greatly acknowledge the contributions of the dropsonde and HALO teams to the mission, are open to future collaboration and understand the difficulties with working around field data, we are confident in our ability to accurately interpret and present the findings without direct involvement from these teams at this point. As experts in the field with a thorough understanding of these type of datasets, we have carefully analyzed and interpreted the data. Additionally, we would like to clarify that the datasets were already publicly available and QA/QC'd when we decided to start the analysis. We trust that any issues with the data have been included as metadata in the files and in the files instructions by the science team, ensuring transparency and reproducibility.*

*As someone who co-led an International Office sponsored by NASA with over 200 partners exchanging data with the public, I learned that open data practices (which NASA follows) should not require someone to contact the science team for basic usage or interpretation. Publicly available datasets are typically accompanied by documentation, metadata, and instructions that provide sufficient information for users to understand and utilize the data effectively. While consultation with the science team may be beneficial for more nuanced or specialized inquiries, it should not be a requirement for accessing, using, or publishing with the data. If a dataset does require users to contact the science team for basic usage or interpretation, it may indicate shortcomings in the dataset's documentation or metadata. In such cases, efforts should be made to improve the accessibility and clarity of the dataset to ensure that users can make full use of the available information without unnecessary barriers or delays. We did not think this was the case here, because again, the data was publicly available at the time.*

*Again, we greatly appreciate all of the effort the Science Team put together and if there are any ways to improve our recognition of specific team members via additional references and or in text comments we would be more than glad to do so. However, these datasets are meant to be used by the public and should not require direct interaction with any science team unless it is really required. Forcing co-authorships from members of the science team limits the dissemination of scientific knowledge to within the science team itself, which runs counter to the spirit of open science. As per NASA's open science policies, while collaboration and consultation with data providers are encouraged and valued, it is not a prerequisite for publication. Our manuscript adheres to NASA's guidelines by ensuring transparency in data usage and methodology.*

*Furthermore, it's worth noting that we consulted with personnel at NASA Headquarters regarding this matter, and our interpretation aligns with their understanding of NASA's open science policies.*

R2: 2. It seems to be an important point in the manuscript that MERRA-2 assimilated aerosol data, but again, it is not explained what data is assimilated or where it's from. Line 124-125 says "The data collected during the CPEX-CV campaign were assimilated into the MERRA-2 reanalysis". Is this really referring to campaign data? The implication is "extinction data from HALO" but I find that very unlikely. MERRA-2 typically assimilates AOT data, from space-based assets and AERONET (I think), but not extinction profiles. Please clarify this (and again, be careful to be specific when talking about observational data to describe it's source). If, as seems likely, MERRA-2 assimilated AOT and yet gets the aerosol profiles incorrect, that would be a good topic of discussion, even if it's not as surprising as the interpretation that I think was implied here, that MERRA-2 assimilates extinction profiles and then fails to come close to reproducing them in the same location and time.

*The assimilation of Lidar data in HALO was mentioned in Nowottnick, Edward P., et al. "The NASA Convective Processes Experiment-Cabo Verde (CPEX-CV): Mission Overview and Saharan Dust Measurements Obtained in the East Atlantic in September 2022." 103rd AMS Annual Meeting. AMS, 2023. A reference was added to the text in line 132.*

In either case, some discussion is needed to explain how assimilation can fail to reproduce the observation data that's assimilated, and how it can fail to reproduce extinction profiles if AOT is assimilated, for readers who may not know the answer to these questions already. Since this point is given some importance in the manuscript, it should be explained fully.

*Divergence between model output and observations in a reanalysis, despite assimilating field mission data, can stem from various sources. These disparities may arise from inherent biases within the model, incomplete correction of model biases during the assimilation process, or uncertainties inherent in observational data. Spatial and temporal mismatches, coupled with differences in resolution between model grids and observational datasets, can also contribute to discrepancies. Furthermore, missing or inadequately represented processes in the model, such as local emissions or atmospheric chemistry, may further exacerbate differences. Additionally, small variations in initial conditions or model parameters can lead to divergent model simulations, even*

*when assimilating the same observational data. Same applies to outputs from different reanalysis. Addressing these discrepancies requires thorough evaluation, sensitivity analysis, and refinement of both model simulations and assimilation techniques which is outside of the scope of this publication.*

R2: 3. Section 2.6.2 gives an equation for heating rate in terms of fluxes, but there is nothing to show how fluxes are obtained from the given dataset. Please give an equation if practical, but more importantly, since software is being used to calculate these, it would be best to see an explicit list of what quantities are needed as input to the software, what quantities are assumed (either in the software or as inputs that are not available from measurements), and what, explicitly are the output quantities. Discuss assumptions, including why they are unavoidable, what precedent is there for the assumed values, what is known about the uncertainty or range for the values, and what is the sensitivity of the results and conclusions to these assumptions.

*The fluxes are obtained from the Fu-Liou-Gu (FLG) radiative transfer mode, and the methodology is explained in section 2.6. The FLG inputs are listed in this section, and we have added a table (Table 1) outlining the input datasets for each run for clarity.*

R2: 4. Design the figures to make the points you want to make about the data clear. For instance, in Figure 3, the most important discussion is about extinction, but the data are almost indistiguishable on the scale given. Perhaps try a logarithmic plot, or in any case, experiment with the presentation of the data until your point is clear.

*We appreciate these suggestions and have adjusted the axis for the 8th and 9th panel in Fig. 3 to make the distinctions more apparent.*

Also the discussion jumps back and forth between Figure 6 and Figure 7 multiple times, making it quite hard to follow because of too much page flipping. Either rearrange the text to match the organization of the figures or vice versa.

*We have updated the discussion of Fig. 6 and 7 to reduce page flipping.*

Most importantly, if the relationship between AEWs and dust is important, there really needs to be a figure conclusively showing that correlation (and adequate discussion to go along with it).

*A figure (Fig. 12), which is discussed above, has been added to the manuscript to illustrate the relationship between AEWs and dust.*

R2: 5. In general, be specific. Instead of "inherent characteristics of the optical properties" (line 299 and repeated word for word at 313), which inherent characteristics?

*Line 299 was meant to apply to LW, not SW, and was corrected accordingly. This line applies to both figures 4 and 5, and thus was moved lower in the text after the discussion of Fig. 5, to the*

*section where the optical properties being referred to (absorption efficiency, scattering) are discussed (lines 324-325 in updated manuscript).*

Instead of "certain aerosols" at 301, which aerosols?

*Examples of aerosols (sulfates, nitrates) that cause LW cooling were added to the text (lines 325-326).*

Most importantly at 331 and at 419, which atmospheric levels are the "crucial atmospheric levels"?

*In line 331, the crucial levels are the ones mentioned in the text leading up to that statement. We have added an explanation of why heating rates matter at these levels in the atmosphere in lines 443-449: "Over the ocean, dust tends to reduce specific humidity in the lower troposphere, particularly in regions with high aerosol loading, while simultaneously augmenting midlevel moisture levels. Dust also warms the lower troposphere, promoting convection and generating positive vorticity between approximately 800–1,000 hPa, where most of the aforementioned SW induced heating rates in both reanalyses prevail. This warming effect can also enhance vertical wind shear. Consequently, this impacts environmental conditions in tropical cyclone genesis regions." Another important factor is that extinction is severely underestimated by the reanalyses around 500 hPa, which affects the heating rates around and below that level. We have clarified which levels we are referring to in the conclusion (line 504).*

R2: 6. In general, be more quantitative. Describe the differences between models and observations quantitatively in the abstract and conclusions. But also, double check all quantitative numbers in the text against the figures. There's at least one spot (line 329) where it looks to be incorrect.

*The abstract and conclusion were updated to include more quantitative numbers, and all quantitative numbers were checked against figures and updated accordingly.*

---

## Author Response (AR2)

[Editor comments are in italics]

Dear Editor,

Thank you for you comments on the revised version of the manuscript. We have addressed these comments and the changes made are outlined below.

*General issues*

*Throughout the manuscript, there are issues with implicitly or explicitly interpreting correlation as causation, without considering or discussing confounding factors, for example (line numbers refer to the tracked manuscript version):*

Response: Thank you for this comment. We understand that there are several places throughout the manuscript where the implications of the text were unclear. To rectify this, we have made several edits to the manuscript that are outlined below.

- *Line 12: "A case study of two developing AEWs highlights a difference in heating rate on the order of 1 to 2 K/day between an AEW developing into a Category 4 Hurricane (Fiona) and a short-lived tropical storm (TS Hermine)."*

Response: To avoid any allusion to a correlation, this sentence in the abstract was changed to "Differences in heating rates were analyzed in a case study of two developing AEWs, one leading to a Category 4 Hurricane (Fiona) and another leading to a short-lived tropical storm (TS Hermine)."

- *Lite 50: "Given that they share similarities in seasonality and geographical extent, the AEWs and Saharan dust are consequently coupled to influence each other."*

Response: This statement was meant to reference results from previous research which was cited later in the paragraph. The text in this paragraph was rearranged to make the connection to the relevant literature more evident, with the citations immediately following the statements.

- *Line 479: "Over the ocean, dust tends to reduce specific humidity in the lower troposphere, particularly in regions with high aerosol loading, while simultaneously augmenting midlevel moisture levels." No physical mechanisms to explain this are discussed.*

Response: A publication (Sun and Zhao, 2020) was cited further down. The text was rewritten to make this citation more evident, and some of the wording in this paragraph was changed to make the conclusions from this statement less ambiguous. We state that while the analysis done here does not allow us to make a conclusion on the impacts of dust on AEW development, "the large differences in heating at the lower levels of the atmosphere between the two cases raise the question of the impact of dust-induced radiative heating on AEW development."

*The manuscript makes strong statements about "truth" without explicitly and quantitatively considering uncertainties:*
*- Although the analysis of heating rates heavily relies on the conversion of backscatter to extinction (depending on size, refractive indices and morphology) and the conversion of these extinction profiles through assumed radiative properties and size (using OPAC) to aerosol radiative properties provided to the radiative transfer model, the resulting uncertainties are not quantitatively discussed. With lidar ratios being under-constrained and OPAC being well-outdated (there exist much better databases on refractive indices these days), this leaves the question how these uncertainties propagate into the calculated heating rate profiles.*

Response:

Thank you for this very relevant comment. We acknowledge that there are more advanced aerosol climatologies available today compared to OPAC. However, OPAC remains widely used in most NWP models which is why it was chosen for this study. The choice was made to ensure consistency with the models that are most commonly used in the community and also both of the reanalysis used in this study also use it. We recognize and share the concerns you bring in here, for example the potential overestimation of single scattering albedo in transported dust in OPAC, and this has been a topic of multiple discussions throughout the course of this experiment. While these issues are important, they are beyond the scope of what we are trying to accomplish in this study.

Regarding the uncertainties associated with the conversion of backscatter to extinction and the use of assumed radiative properties and sizes from OPAC, it is important to note that these conversions are performed by the science team responsible for the data and are included in the dataset as they are. As users, we have assumed that these conversions are correct and have been done to the best possible knowledge of the science team. However, we recognize that these assumptions may introduce uncertainties, particularly in how they might propagate into the calculated heating rate profiles. Previous studies have highlighted similar uncertainties. For example, Dubovik

et al. (2002) examine the influence of particle nonsphericity on the retrieval of aerosol optical properties, while Kahn et al. (2005) discuss the uncertainties in aerosol models derived from satellite data. Additionally, Levy et al. (2010) address the challenges in retrieving accurate aerosol optical properties and their implications for climate models.These studies suggest that the propagation of such uncertainties can significantly affect the accuracy of radiative transfer models.

- *It is therefore not clear how to interpret the presented differences between observationally derived heating rates (referred to as "truth") and reanalyses – do they lie within or without the error bars for derived heating rates?*

Response: In response to your concerns, we have made some edits to the wording in *Section 3* and have added lines in *Section 2.6.1 OPAC* to address these issues. The word "truth" was replaced with "observation" or "observational data" in *Section 3.2 Impact of aerosol on heating rates* and in *Section 3.3 Dataset comparison*, and a paragraph on the limitations of OPAC was added to the conclusions. While a detailed quantitative analysis of these uncertainties was beyond the scope of this paper, we acknowledge their significance and suggest this as an important area for future research. In future studies, incorporating more recent databases on refractive indices and considering the implications of under-constrained lidar ratios could provide a more refined analysis.

*Specific issues:*
- *Title: consider adding "the" ahead of AEW Development Region"*

Response: The title was changed accordingly.

- *Fig. 1: "Image" is not reproducible for an imager with many channels.*

Response: The word "image" was removed from the figure caption.

- *Line 165: AEW and storms are related but seem to be used here interchangeably. This requires an appropriate definition of terminology and consistent use throughout.*

Response: We appreciate this comment and agree that the distinction needs to be made clearer. In *Section 2.4 AEW Tracking*, the word storm was changed to AEW. In *Section 3.1 Description of AEW events during CPEX-CV*, the word storm was qualified to clarify whether it was referring to an AEW or a tropical storm. *In Section 3.5 Case study: Hurricane Fiona and TS Hermine*, the word "storm" is replaced in several

locations depending on what it is referring to with: "AEW (and the TC it develops into)", "AEW/TC", "development of the AEW" and "TS", and removed completely in one case. In *Section 4 Conclusion*, the word storm was changed to "TS" in one case and to "development of the AEW" in another.

- *Line 296: This statement is confusing. HALO is flown on CPEX-CV and shows the variability. Presumably you mean something different here?*

Response: This typo was changed from "CPEX-CV" to "CAMS".

Other changes made:
Typos were corrected throughout the text. Some repetitions in the methods section were removed. Minor edits were also made for clarity.